# The TUTase URT1 connects decapping activators and prevents the accumulation of excessively deadenylated mRNAs to avoid siRNA biogenesis

Hélène Scheer[1,5], Caroline de Almeida[1,5], Emilie Ferrier[1], Quentin Simonnot[1], Laure Poirier[1], David Pflieger[1], François M. Sement[1], Sandrine Koechler[1], Christina Piermaria [2], Paweł Krawczyk [3,4], Seweryn Mroczek [3,4], Johana Chicher [2], Lauriane Kuhn [2], Andrzej Dziembowski[3,4], Philippe Hammann[2], Hélène Zuber [1✉] & Dominique Gagliardi [1✉]

Uridylation is a widespread modification destabilizing eukaryotic mRNAs. Yet, molecular mechanisms underlying TUTase-mediated mRNA degradation remain mostly unresolved. Here, we report that the Arabidopsis TUTase URT1 participates in a molecular network connecting several translational repressors/decapping activators. URT1 directly interacts with DECAPPING 5 (DCP5), the Arabidopsis ortholog of human LSM14 and yeast Scd6, and this interaction connects URT1 to additional decay factors like DDX6/Dhh1-like RNA helicases. Nanopore direct RNA sequencing reveals a global role of URT1 in shaping poly(A) tail length, notably by preventing the accumulation of excessively deadenylated mRNAs. Based on in vitro and in planta data, we propose a model that explains how URT1 could reduce the accumulation of oligo(A)-tailed mRNAs both by favoring their degradation and because 3′ terminal uridines intrinsically hinder deadenylation. Importantly, preventing the accumulation of excessively deadenylated mRNAs avoids the biogenesis of illegitimate siRNAs that silence endogenous mRNAs and perturb Arabidopsis growth and development.

[1] Institut de biologie moléculaire des plantes, CNRS, Université de Strasbourg, Strasbourg, France. [2] Plateforme Protéomique Strasbourg Esplanade du CNRS, Université de Strasbourg, Strasbourg, France. [3] Laboratory of RNA Biology, International Institute of Molecular and Cell Biology, Warsaw, Poland. [4] Faculty of Biology, Institute of Genetics and Biotechnology, University of Warsaw, Warsaw, Poland. [5] These authors contributed equally: Hélène Scheer, Caroline de Almeida. ✉email: helene.zuber@ibmp-cnrs.unistra.fr; dominique.gagliardi@ibmp-cnrs.unistra.fr

Uridylation targets most classes of eukaryotic RNAs, from small and large noncoding RNAs (ncRNAs) to mRNAs[1–3]. Uridylation of ncRNAs can promote maturation, control stability, or abrogate activity, depending on the type of ncRNAs and its cellular context[2–4]. For mRNAs, the prevalent role of uridylation is to trigger 5′-3′ and 3′-5′ degradation[5–8].

mRNA uridylation is preceded by deadenylation[5,6,9–12], which is mostly achieved by the multifunctional Carbon Catabolite Repression-Negative On TATA-less (CCR4-NOT) complex[13]. Two of its core subunits are the deadenylases CCR4 and CCR4-associated factor 1 (CAF1)[13,14]. CCR4 is proposed to shorten poly(A) tails of all mRNAs while CAF1 deadenylates mRNAs with lower rates of translation elongation and poor poly(A) binding protein (PABP) occupancy[15]. Importantly, specificity factors bridge target mRNAs to the CCR4-NOT complex, thereby leading to translational repression and decay of specific mRNAs. Those specificity factors include the RNA-induced silencing complex (RISC) and several RNA binding proteins (RBPs), such as Tristetraprolin (TTP), Pumilio/fem-3 mRNA binding factor (PUF) proteins, and the YT521-B homology (YTH) domain-containing family proteins YTHDF2 and Meiotic mRNA interception protein 1 (Mmi1)[13,16–19]. In addition, the CCR4-NOT complex interacts with central regulators of translation and decapping, such as the GRB10-interacting GYF (glycine-tyrosine-phenylalanine domain) proteins (GIGYF)[20,21] and the DExD/H-box RNA helicases DDX6, Dhh1, or Maternal expression at 31B (Me31B) in humans, yeast, and *Drosophila*, respectively[13]. DDX6 also interacts with the decapping activators Enhancer of mRNA-decapping protein 3 (EDC3), DNA topoisomerase 2-associated protein (PAT1), and Like SM homolog 14 (LSM14), also called Suppressor of clathrin deficiency (Scd6) in yeast. The binding of DDX6 to CNOT1 or to decapping activators is proposed to be mutually exclusive. A possible scenario is that a succession of interactions allows DDX6 to hand over deadenylated mRNAs to the decapping machinery[22].

Once the CCR4-NOT complex has been recruited and the poly(A) tail has been shortened, oligo(A) tails of less than ca 25 As are frequently uridylated by terminal uridylyltransferases (TUTases) such as TUT4/7 in mammals or UTP:RNA URIDYLYLTRANSFERASE (URT1) in *Arabidopsis thaliana* (hereafter Arabidopsis)[6,11]. URT1-mediated uridylation can restore a binding site for a PABP, but its impact on mRNA stability is yet unsolved[1,4,11,12]. In fission yeast and human cultured cells, uridylation of short oligo(A) tails is proposed to favor the binding of the LSm1-7 complex, which recruits the decapping complex through the interaction with PAT1, and ultimately results in the degradation of the decapped mRNA by the 5′-3′ exoribonuclease 1 XRN1 (XRN4 in plants)[5,6,8,23]. Alternatively, U-tails can directly attract Dis3-like protein 2 (Dis3L2) or the RNA exosome to trigger 3′-5′ exoribonucleolytic decay of mRNAs[6,24].

In this study, we show that the Arabidopsis TUTase URT1 is integrated in an interaction network comprising the deadenylation complex CCR4-NOT and other translation repressors/decapping activators, including DCP5, the plant ortholog of the translational inhibitor/decapping activator Scd6 or LSM14A. Our interactomic and functional analysis data support a model explaining how URT1-mediated uridylation prevents the accumulation of excessively deadenylated mRNAs. We also show that in absence of URT1-mediated uridylation, excessively deadenylated mRNAs can become a source of spurious siRNAs that silence endogenous mRNAs, with a negative impact on plant fitness.

## Results

### Conservation of an intrinsically disordered region (IDR) across plant URT1 orthologs.
The 764-long amino acid sequence of the

Arabidopsis TUTase URT1 encoded by *AT2G45620* can be divided in two regions discriminated by compositional biases and the presence of known domains (Fig. 1a). URT1's C-terminal region contains the catalytic core domain (CCD), the typical signature of terminal nucleotidyltransferase family members. The CCD is composed of a nucleotidyltransferase domain (amino acids 434–567, Superfamily domain SCOP 81302, $E$ value $= 3.11 \times 10^{-36}$) followed by a PAP-associated domain or PAP/OAS1 substrate-binding domain (amino acids 571–741, Superfamily domain SCOP 81631, $E$ value $= 6.8 \times 10^{-48}$) (Fig. 1a). By contrast, the N-terminal region of URT1 (amino acids 1–433) is devoid of known domains and is characterized by a significant enrichment for P/Q/N/G ($p$ value $= 2.4 \times 10^{-18}$) compared to the Arabidopsis proteome (Fig. 1a). Moreover, the whole N-terminal half of URT1 is predicted as a large IDR (Fig. 1b).

To test for the possible conservation of this IDR amongst plant URT1 orthologs, we analyzed 87 sequences (Supplementary Data 1) that were recently compiled to determine the evolutionary history of TUTases in Archaeplastida (i.e., all plants)[1]. These URT1 sequences originate from 73 species representing major groups of Archaeplastida: glaucophytes, rhodophytes (red algae), chlorophyte and streptophyte algae, bryophytes (liverworts, hornworts, and mosses), lycophytes and pteridophytes (e.g., ferns), gymnosperms (e.g., conifers and Gingko), and angiosperms (flowering plants)[1]. In all groups, we identified URT1 orthologs with a predicted N-terminal IDR (Fig. 1c). The preservation of this IDR throughout the plant cell lineage likely indicates a conserved key function.

### Short linear motifs (SLiMs) are conserved in plant URT1 orthologs.
IDRs can tolerate mutations that do not affect their overall function. Indeed, the primary sequence of URT1's IDR is highly variable between species (Fig. 1d and Supplementary Fig. 1a). Yet, two SLiMs named hereafter M1 and M2 are conserved in land plants, i.e., from bryophytes (including mosses) to flowering plants (Fig. 1d, e and Supplementary Fig. 1a). A PPGF motif is also conserved in many land plant URT1s (Supplementary Fig. 1a). Its conservation is underestimated by sequence alignment partly because of its varying positions in the IDR (Fig. 1e). Yet, a systematic search revealed that URT1 orthologs of most flowering plants and several mosses contain a PPGF motif (Fig. 1e). Poales, which are monocotyledons that include key cereal crops, are among the plants whose genome encodes two URT1 homologous sequences, URT1A and URT1B[1]. URT1A isoforms are ubiquitously expressed, as is URT1, whereas the expression of *URT1B* genes is often restricted to specific tissues[1]. Interestingly, URT1As contain the M1, M2, and PPGF motifs whereas URT1B sequences contain only a divergent M1 and lack both the M2 and PPGF motifs (Fig. 1e and Supplementary Fig. 1b). Altogether, these observations suggest a potential specialization of URT1A and URT1B. Conversely, the conservation in flowering plants of at least one URT1 ortholog with M1, M2, and PPGF motifs indicates key functions under selective pressure.

### URT1 co-purifies with translational repressors/decapping activators.
The conservation of a large IDR containing SLiMs supports the possibility that URT1 interacts with one or several partners. To identify this interaction network, proteins co-purifying with URT1 (tagged with myc or YFP) expressed in *urt1* mutants were identified by LC-MS/MS analyses. To obtain a global view of URT1 RNP context, cellular extracts were cross-linked with formaldehyde before immunoprecipitation (IP) (Fig. 2a). The comparison of eight URT1 samples (representing four biological replicates) to seven control samples revealed 62 proteins significantly enriched in URT1 IPs (Fig. 2a

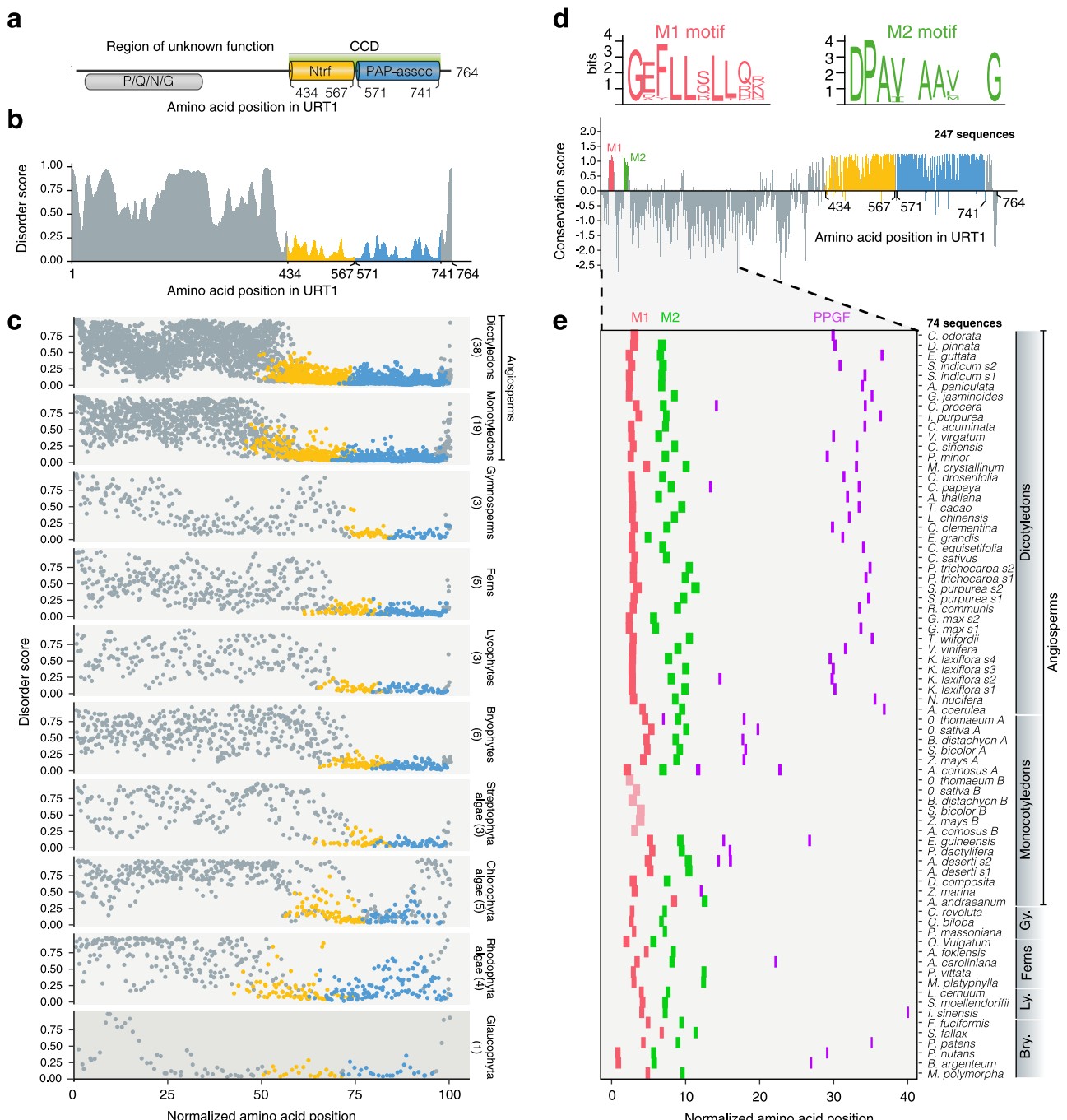

**Fig. 1 Short linear motifs are conserved in plant URT1 orthologs. a** Domain organization of URT1. CCD catalytic core domain, Ntrf polymerase β-like nucleotidyltransferase domain, PAP-assoc Poly(A) polymerase-associated domain. The region of P/Q/N/G enrichment is indicated. Numbers refer to amino acid positions in URT1. **b** Disorder propensity of URT1 predicted by ESpritz-NMR[74] and aggregated by FELLS[75]. **c** Disorder propensity of URT1 orthologs in Archaeplastida. The length of URT1 sequences was normalized to 100. The number of sequences considered are indicated for each group. **d** Sequence conservation of URT1 orthologs among land plants. The conservation score was calculated with ConSurf[76] from an alignment of 247 sequence homologs of URT1 in land plants (see Supplementary Data 1) and using URT1 sequence as a reference. Consensus logos for M1 and M2 motifs are displayed in bits and were calculated from the alignment. **e** Occurrence of M1 (red), M2 (green), and PPGF (purple) motifs among URT1 orthologs of land plants. The slightly divergent M1 motif of URT1B in Poales is in light red. The length of 74 URT1 sequences was normalized to 100. The source data are available in Supplementary Data 1.

and Supplementary Data 2). The most enriched molecular functions associated to URT1 co-purifying proteins are mRNA binding (GO:0003729) and RNA binding (GO:0003723) (Benjamini–Hochberg corrected $p$ values of $4.8 \times 10^{-29}$ and $6.1 \times 10^{-18}$, respectively). Both categories are consistent with the known involvement of URT1 in mRNA metabolism[11,12].

The most significantly enriched protein co-purifying with URT1 is ESSENTIAL FOR POTEXVIRUS ACCUMULATION 1 (EXA1) encoded by *AT5G42950* and also named GYN4, PSIG1, and MUSE11[25–28]. The two closest EXA1 homologs (*AT1G27430* and *AT1G24300*) are also significantly enriched in URT1 IPs (noted GYF protein in Fig. 2a and Supplementary Data 2a). EXA1

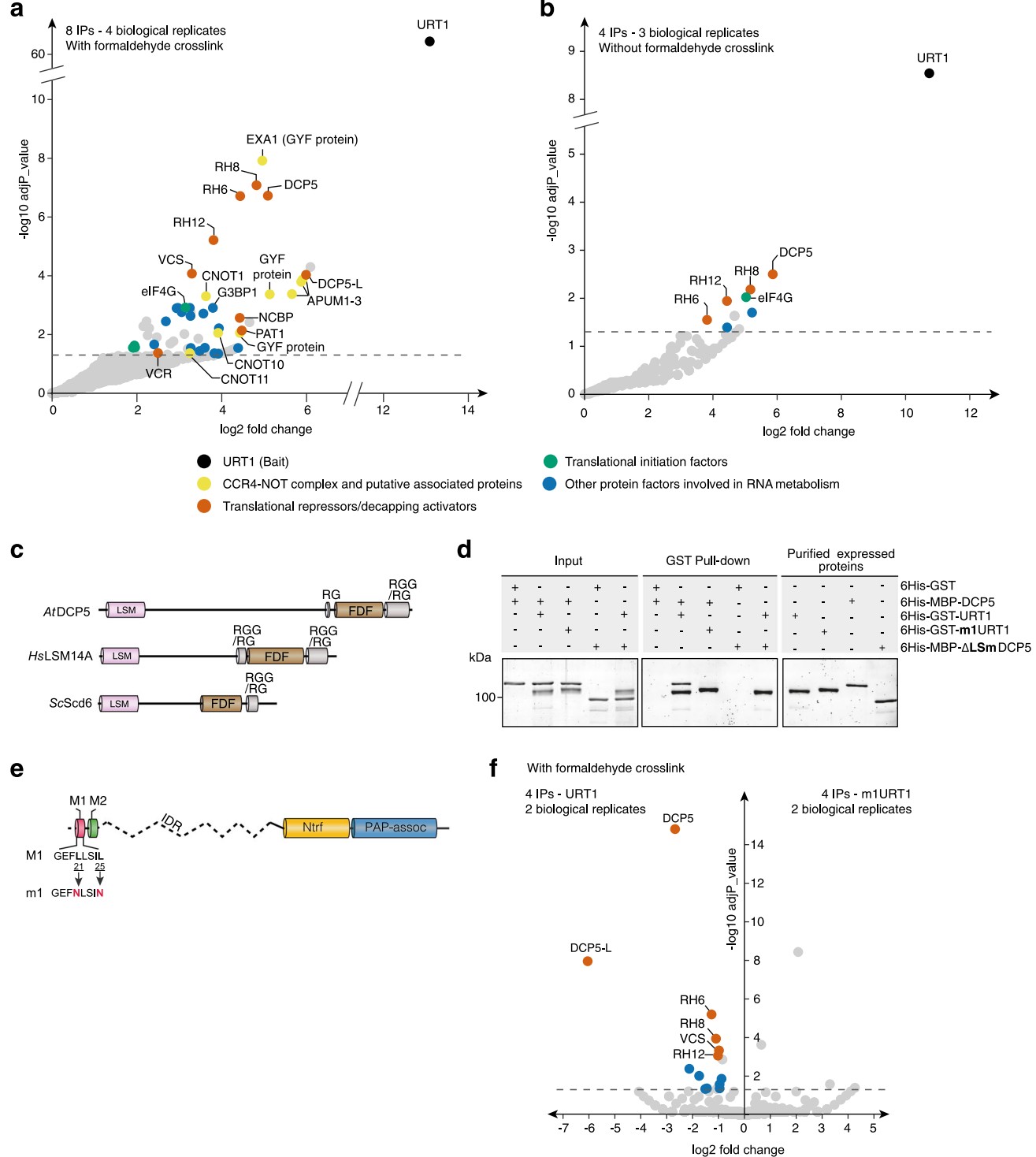

**Fig. 2 URT1 co-purifies with translational repressors/decapping activators.** Enrichment of proteins co-purified with myc and YFP-tagged URT1 with formaldehyde crosslink (**a**) or without (**b**). The dashed line indicates the threshold above which proteins are significantly enriched (adjusted *p* value < 0.05, quasi-likelihood negative binomial generalized log-linear model). **c** Common domain organization of Arabidopsis (*At*) DCP5 and its human (*Hs*) and yeast (*Sc*) orthologs LSM14A and Scd6, respectively. **d** In vitro GST pull-down assay showing a direct URT1-DCP5 interaction. Pull-downs were performed in presence of RNase A with the recombinant proteins 6His-GST, 6His-GST-URT1, 6His-MBP-DCP5, 6His-GST-m1URT1, and 6His-MBP-ΔLSmDCP5. Pull-down and input fractions were analyzed by SDS-PAGE and SYPRO Ruby staining. **e** Diagram illustrating the point mutations in m1URT1 construct. **f** Volcano plot showing proteins differentially enriched (log2 fold change > 0) or depleted (log2 fold change < 0) in myc-tagged m1URT1 versus myc-tagged URT1 IPs. The dashed line indicates the significant threshold (adjusted *p* value < 0.05, quasi-likelihood negative binomial generalized log-linear model). Dot color code is as in **a**. The source data are available in Supplementary Data 2, at [https://www.ebi.ac.uk/pride/archive/projects/PXD018672] and at [https://doi.org/10.17632/ybcvvmtcn9.3].

is a GYF domain-containing protein, orthologous to human and Drosophila GRB10-interacting GYF domain proteins (GIGYF). GIGYF proteins interact with the deadenylation complex CCR4-NOT and several translational repressors or decapping activators such as the 5′ cap-binding protein eIF4E-Homologous Protein (4EHP), the RNA helicase DDX6/Me31B, and the decapping activator PAT1[20,21]. Interestingly, orthologs of all known GIGYF interactors are also significantly enriched in URT1 IPs alongside EXA1. They include subunits of the CCR4-NOT complex like CNOT1, CNOT10, and CNOT11 (AT1G02080, AT5G35430, and AT5G18420, respectively), the 4EHP ortholog called new cap-binding protein (nCBP) (AT5G18110), PAT1 (AT1G79090) and the DDX6-like RNA helicases RH6, RH8, and RH12 (AT2G45810, AT4G00660, and AT3G61240, respectively). These results raise the possibility that the chain of interactions described for GIGYF is conserved for EXA1 in Arabidopsis and that URT1 is connected to these factors, including the CCR4-NOT complex.

Another translational repressor/decapping activator highly enriched in URT1 IPs is DECAPPING5 (DCP5) encoded by AT1G26110 (Fig. 2a and Supplementary Data 2a). The known interactants of the human DCP5 ortholog LSM14 are EDC4, the DDX6 RNA helicases, and the eIF4E-binding protein 4E-T[29]. There is no 4E-T ortholog in Arabidopsis. However, both VARICOSE (VCS), ortholog to EDC4 and encoded by AT3G13300, and the aforementioned DDX6-like RNA helicases (RH6, RH8, and RH12) are enriched in URT1 IPs. Of note, DECAPPING 5-LIKE (DCP5-L, AT5G45330) and VARICOSE-RELATED (VCR, AT3G13290), homologs of DCP5 and VCS, respectively, are also significantly enriched in URT1 IPs (Fig. 2a and Supplementary Data 2).

Repeating the IP experiments without formaldehyde crosslink revealed a much simpler interactome with DCP5, RH6, RH8, RH12, and the translation initiation factor eIF4G (AT3G60240) among the most enriched proteins (Fig. 2b and Supplementary Data 2b). A DCP5-eIF4G interaction has not yet been reported, but the yeast DCP5 ortholog Scd6 does interact with eIF4G via its C-terminal RGG repeats[30], which are also present in DCP5 and LSM14A (Fig. 2c).

Altogether, our IP results indicate that URT1 is integrated into interaction networks connecting translational repressors and decapping activators, including DCP5.

**The SLiM M1 mediates a direct interaction between DCP5 and URT1.** DCP5 is the most enriched protein in mycURT1 IPs without crosslinking (Fig. 2b and Supplementary Data 2b) and reciprocally, URT1 co-purifies with DCP5 tagged with GFP and expressed at endogenous levels (Supplementary Fig. 2a). DCP5 contains a LSm domain, and the LSm domain-containing proteins LSM14, Scd6, and Edc3 interact with helical leucine-rich motifs (HLMs)[31], which resemble URT1's M1 motif (Fig.1d). We therefore suspected a direct DCP5-URT1 interaction. Indeed, in vitro pull-down experiments in presence of RNase A confirmed a direct interaction between 6His-GST-URT1 and 6His-MBP-DCP5 (Fig. 2d). This direct interaction requires the M1 motif because a mutated version of URT1, in which leucines 21 and 25 in M1 are mutated into asparagines (6His-GST-m1URT1) (Fig. 2e) failed to pull down DCP5 (Fig. 2d). Furthermore, 6His-GST-URT1 cannot pull down 6His-MBP-ΔLSmDCP5, indicating that the LSm domain of DCP5 is necessary for the URT1-DCP5 interaction (Fig. 2d). Additional assays confirmed that 6His-GST-URT1[1−40] (i.e., the 40 first aminoacids of URT1 containing the M1 motif but not the M2 motif and fused downstream of 6 histidines and GST) is sufficient to pull down 6His-MBP-DCP5[LSm] (i.e., DCP5's LSm domain fused downstream of 6 histidines and MBP). As expected, this interaction is abolished by

mutating the M1 motif (Supplementary Fig. 2b). By contrast, mutating the M2 motif has no impact on the URT1-DCP5 pull-down assays (Supplementary Fig. 2c). Altogether, these data reveal that URT1 can directly bind to DCP5 via an interaction between URT1's conserved M1 motif and DCP5's LSm domain.

To test how M1 impacts the URT1 interactome in planta, URT1-myc or m1URT1-myc were expressed in urt1-1 mutant plants and used as baits in IPs following formaldehyde crosslink. Interestingly, the six proteins most significantly depleted by mutating the M1 motif are DCP5, DCP5L, RH6, RH8, VCS, and RH12 (Fig. 2f and Supplementary Data 2c). We conclude from these experiments that the conserved SLiM M1 connects URT1 to DCP5 (and possibly DCP5L), which recruits additional translation repressors or decapping activators such as VCS or the Dhh1/DDX6-like RNA helicases, RH6, RH8, and RH12.

**Ectopic expression of URT1 remodels poly(A) tail profiles.** To investigate the molecular function of URT1-mediated mRNA uridylation and test the potential role of the M1 and M2 motifs, we first determined how the ectopic expression of URT1-myc or m1m2URT1-myc affects the expression of a GFP reporter mRNA co-expressed in Nicotiana benthamiana leaves. To prevent transgene-induced silencing, the silencing suppressor P19 was co-expressed in all experiments and will not be further mentioned. The GFP reporter was co-expressed without URT1 (ctrl) or with either one of two catalytically impaired versions of URT1, URT1[D491/3A], or URT1[P618L] (see Fig. 3a for a schematic representation of the different URT1 versions). URT1[D491/3A] is fully inactivated by the mutations of catalytic residues[11] whereas the uridylating activity of URT1[P618L] mutant is strongly affected, but not abrogated[32].

Both active and inactive URT1 versions were expressed as full-length proteins (Fig. 3b). Yet, the inactive versions of URT1 are systematically more expressed as compared with active ones, which is important for the interpretation of the results presented hereafter. GFP expression levels were monitored by UV illumination of infiltrated leaf patches. Leaf patches expressing catalytic inactive versions of URT1 showed GFP levels similar to controls. By contrast, the GFP fluorescence was systematically decreased upon expression of active versions of URT1 with either wild-type or mutated M1 and M2 motifs (Fig. 3c). Thus, GFP repression requires URT1's activity but obviously neither the M1 nor the M2 motif. We do not know at present whether GFP repression is a direct consequence of GFP mRNA uridylation, or due to an indirect effect of URT1 ectopic overexpression. Of note, GFP repression is not the consequence of a lower expression of the silencing suppressor P19 because similar results were obtained with N. benthamiana plants silenced for RDR6, a key component of transgene-induced post-transcriptional silencing (PTGS) (Supplementary Fig. 3a). Remarkably, the decrease in GFP expression was not due to lower amounts of GFP mRNAs, because similar steady-state levels of GFP mRNAs were detected in patches expressing active or inactive URT1 (Fig. 3d). Yet, expression of the inactive URT1[D491/3A] resulted in a slight, but systematic, shift in the migration of GFP mRNAs detected by northern blots (Fig. 3d). To investigate the reason for this small size shift and to determine the molecular impact of ectopic URT1 expression on GFP mRNA tails, those mRNAs were investigated by 3′RACE-seq. A primer was ligated to RNA 3′ extremities and used to initiate cDNA synthesis. The 3′ region of GFP reporter mRNAs (including the poly(A) tail and eventually non-A nucleotides) was PCR-amplified and GFP amplicons from six independent biological replicates were sequenced in two MiSeq runs (Supplementary Data 3a, b). In control patches, GFP mRNAs had a mean uridylation level of 3.2% and uridylation

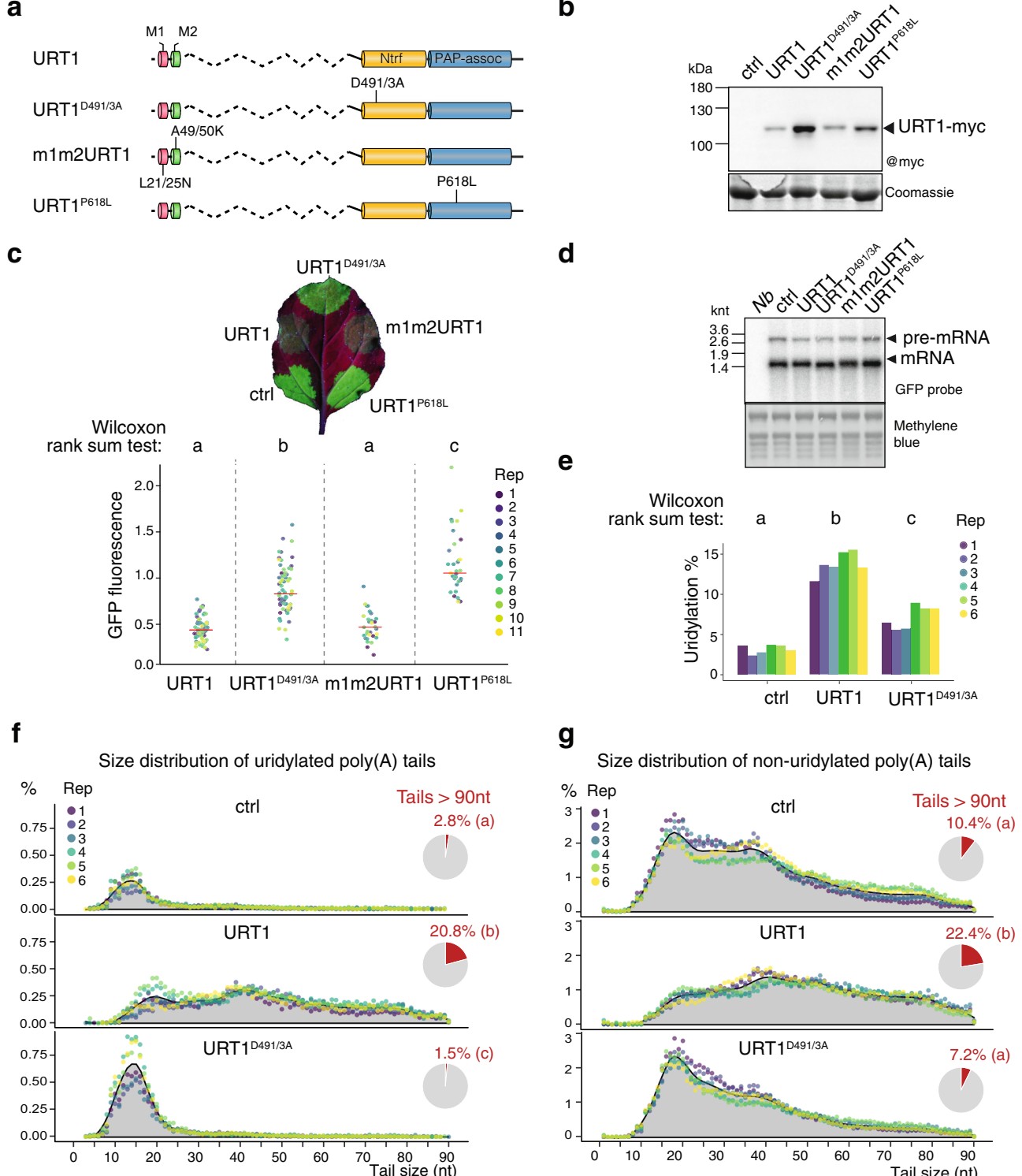

**Fig. 3 Ectopic expression of URT1 remodels poly(A) tail profiles. a** Domain organization and mutations of different URT1-myc versions transiently expressed in *N. benthamiana* leaf patches. Legend as in Fig. 1a. **b** Western blot analysis of URT1-myc expression. **c** Representative *N. benthamiana* leaf under UV light to detect the expression of the GFP reporter co-expressed with the different URT1-myc versions (top). Quantification of GFP fluorescence of the different patches relative to control (ctrl) for 11 independent replicates (bottom). **d** Northern blot analysis of the steady-state level of *GFP* mRNAs. The arrows indicate unspliced and mature forms of *GFP* mRNAs. **e** Uridylation percentage of *GFP* reporter mRNAs for six biological replicates. **f, g** Distribution profiles of *GFP* mRNA poly(A) tail sizes. The percentages of sequences were calculated for six biological replicates for uridylated (**f**) and homopolymeric poly(A) tails (non-uridylated) (**g**) tails from 1 to 90 nucleotides. The percentages were calculated using the total number of sequences with tails from 1 to 90 nucleotides. Individual points are color-coded for each replicate and the average of all replicates is indicated as a gray area. The pie charts represent the average proportion of tails longer than 90 nucleotides. Letters in (**c**, **e–g**) represent significant statistical *p* value (two-tailed Wilcoxon rank-sum test, *n* = 11 (**c**) and *n* = 6 (**e–g**)). Exact *p* values are indicated in Supplementary Data 3e. The source data are available in Supplementary Data 3, at [https://www.ncbi.nlm.nih.gov/geo/query/acc.cgi?acc=GSE148409] and at [https://doi.org/10.17632/ybcvvmtcn9.3].

occurred mostly on *GFP* mRNAs with 10-25 As (ctrl in Fig. 3e, f). This mRNA population with oligo(A) tails of 10–25 As will be later referred to as oligoadenylated mRNAs. The *GFP* mRNA uridylation pattern resembles the ones typically observed for *Arabidopsis* mRNAs[10–12] and we propose that this basal uridylation level is performed by the *N. benthamiana* URT1 ortholog. As expected, *GFP* mRNA uridylation levels increased upon ectopic expression of URT1-myc and reached up to 15% (Fig. 3e). Interestingly, the size distribution profile for uridylated tails was markedly modified by the ectopic expression of URT1-myc, which resulted in the uridylation of large poly(A) tails up to 90 As (Fig. 3f). Moreover, the number of uridylated tails longer than 90 As was also significantly higher upon ectopic expression of URT1 as compared to the control samples or leaf patches expressing inactive URT1 (pie charts in Fig. 3f). Thus, URT1 can uridylate long poly(A) tails when ectopically expressed, demonstrating that deadenylation is not a pre-requisite for uridylation.

Strikingly, URT1 ectopic expression also affected the size distribution of homopolymeric poly(A) tails (called non-uridylated poly(A) tails hereafter and in all figures). Indeed, non-uridylated poly(A) tails in samples expressing wild-type URT1 showed a clear decrease of the 16–20 peak and an accumulation of larger poly(A) tails as compared to control samples (Fig. 3g and Supplementary Fig. 3b). This accumulation is due to the uridylation activity of URT1, as it is not observed for URT1$^{D491/3A}$ (Fig. 3g), despite the higher expression levels of the inactive protein (Fig. 3b).

Another somehow counterintuitive observation was that overexpression of inactive URT1$^{D491/3A}$ resulted in increased *GFP* mRNA uridylation as compared to control samples (Fig. 3e). This increased uridylation corresponds to the specific over-accumulation of uridylated oligoadenylated *GFP* mRNAs as compared to control samples (Fig. 3f). We hypothesized that inactive URT1$^{D491/3A}$ overexpressed to high levels (Fig. 3b) may sequester decay factors, thereby impeding the degradation of uridylated oligoadenylated mRNAs. The resolution of the URT1 interactome in *Arabidopsis* hinted that the M1 motif could participate in sequestering decay factors. Therefore, we tested whether the accumulation of oligoadenylated *GFP* mRNAs upon ectopic overexpression of URT1$^{D491/3A}$ requires the M1 motif. To do so, we co-expressed the reporter *GFP* mRNAs with URT1, URT1$^{D491/3A}$, m1URT1$^{D491/3A}$, m2URT1$^{D491/3A}$, or m1m2URT1$^{D491/3A}$ (see Fig. 4a for a schematic representation of the different URT1 versions). As previously noted, URT1 inactive versions were expressed more highly than active URT1 (Fig. 4b). We then analyzed *GFP* mRNA tailing profiles by 3′ RACE-seq. In line with our previous results, the ectopic expression of URT1 resulted in much longer poly(A) tails for both uridylated and non-uridylated *GFP* mRNAs (Fig. 4c, d, respectively), and uridylated oligoadenylated *GFP* mRNAs accumulated upon URT1$^{D491/3A}$ expression (Fig. 4c). Interestingly, the accumulation of uridylated oligoadenylated *GFP* mRNAs was reduced by mutating the M1, but not the M2 motif (Fig. 4c). This observation supports the idea that overexpressed URT1$^{D491/3A}$ sequesters a factor(s) involved in the turnover of uridylated oligoadenylated mRNAs through an interaction involving the M1 motif. Indeed, the comparison of MS data for seven URT1$^{D491/3A}$ versus eight m1-URT1$^{D491/3A}$ IPs revealed that URT1$^{D491/3A}$ co-purifies with *N. benthamiana* decay factors including DCP5 homologs and that this interaction requires the M1 motif (Supplementary Fig. 3c, d).

Finally, we noted that poly(A) tails interspersed with non-A ribonucleotides, subsequently called A-rich tails, accumulated in URT1$^{D491/3A}$ samples. This accumulation is strictly dependent on the presence of the M1 motif in overexpressed URT1$^{D491/3A}$ (Fig. 4e and Supplementary Fig. 3e). Of note, only few A-rich tails

are detected in control or URT1 samples, and they could have been considered as possible experimental artefacts. However, their dramatic accumulation in URT1$^{D491/3A}$ provides compelling evidence that these A-rich tails are produced in vivo. Either A-rich tails are constitutively produced but do not accumulate in wild-type plants, or their production is induced by overexpressing M1-containing URT1$^{D491/3A}$. Their simultaneous accumulation with uridylated and oligoadenylated mRNAs, triggered by overexpressing URT1$^{D491/3A}$ versions that contains the M1 motif, suggests that these tails mark mRNAs undergoing degradation.

Taken altogether, those data confirm that URT1 connects decay factors through the M1 motif and, more importantly, reveal that the ectopic overexpression of URT1 remodels the poly(A) tails of *GFP* reporter mRNAs toward larger sizes.

**Effects of URT1 ectopic expression on tailing of endogenous *PR2* mRNAs.** We abstained from tethering URT1 to the *GFP* reporter mRNA because URT1 is a distributive TUTase[11] and tethering would likely entail the synthesis of longer poly(U) tails as compared to the wild-type situation. In our experimental design, the number of uridines added to *GFP* mRNAs is similar between control and URT1 samples (Supplementary Fig. 3f). Because ectopically expressed URT1 is not tethered to the reporter mRNAs, it potentially uridylates also endogenous mRNAs. To test this possibility, endogenous mRNAs encoding PATHOGENESIS-RELATED PROTEIN 2 (PR2), were analyzed by 3′RACE-seq. *PR2* mRNAs were chosen because the agroin-filtration procedure triggers *PR2* expression (Supplementary Fig. 4a). Unlike for *GFP* mRNAs, overexpression of URT1$^{D491/3A}$ led to increased levels of *PR2* mRNAs (Supplementary Fig. 4a). Either overexpression of URT1$^{D491/3A}$ induces *PR2* transcription at a higher level, or URT1$^{D491/3A}$ overexpression impairs *PR2* mRNA turnover. In line with the *GFP* mRNA results, URT1 ectopic expression increased *PR2* uridylation levels, resulted in the uridylation of longer poly(A) tails, and led to the accumulation of *PR2* mRNAs with longer poly(A) tails as compared to the control samples (Supplementary Fig. 4b–d). Moreover, URT1$^{D491/3A}$ overexpression led to the accumulation of oligo-denylated uridylated *PR2* mRNAs, as well as to the accumulation of *PR2* mRNAs with A-rich tails (Supplementary Fig. 4b, c, e). Altogether, these data indicate that URT1 ectopic expression has a similar impact on tail sizes on both the reporter *GFP* mRNAs and the endogenous *PR2* mRNAs.

**URT1-mediated uridylation shapes poly(A) tails in Arabidopsis.** If the shift toward larger poly(A) tail sizes induced by URT1 overexpression reflects bona fide molecular functions of this TUTase, the opposite effect should be observed in *urt1* mutants. We therefore compared global poly(A) tail profiles between three biological replicates of wild-type and *urt1* plants using nanopore direct RNA sequencing (DRS). Although DRS does not yet allow to detect mRNA uridylation, it is well suited for measuring poly(A) tail sizes, including long ones. The nanopore DRS analysis revealed the impact of URT1 on poly(A) tail size distributions with a clear shift of the distribution toward short oligo(A) tails in *urt1* samples (Fig. 5a). This accumulation of deadenylated mRNAs is observed for both bulk and intergenic poly(A) tail size distributions, thereby reflecting a robust effect.

We then used the 3′RACE-seq method to precisely compare polyadenylation and uridylation profiles for 22 mRNAs analyzed in two biological replicates of wild-type and *urt1* plants. Those mRNAs were selected because they have various uridylation levels ranging from 1 to 24 % (Fig. 5b), and quite distinct poly(A) tail profiles (for instance compare *AT2G30570* and *AT4G28240*, the

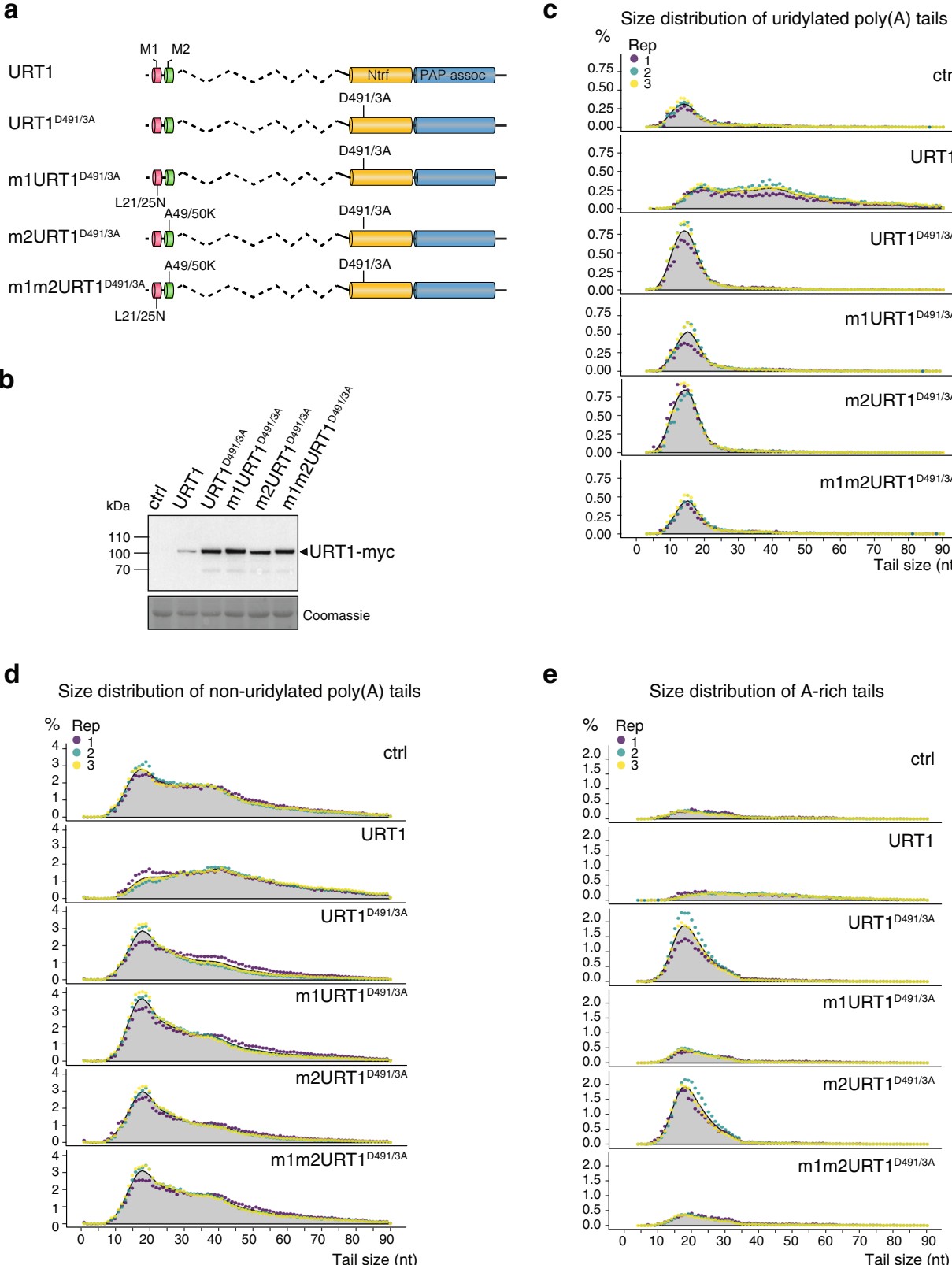

**Fig. 4 Accumulation of oligo(A) uridylated and A-rich-tailed *GFP* mRNAs upon overexpression of URT1$^{D491/3A}$ requires the M1 motif. a** Domain organization and mutations of URT1-myc versions. **b** Western blot analysis of URT1-myc expression. **c–e** Distribution profiles of *GFP* mRNA poly(A) tail sizes. The percentages of sequences were calculated for three biological replicates for uridylated (**c**), non-uridylated (**d**) and A-rich (**e**) tails from 1 to 90 nucleotides. The percentages were calculated using the total number of sequences with tails from 1 to 90 nucleotides. Individual points are color-coded for each replicate and the average of all replicates is indicated as a gray area. The source data are available in Supplementary Data 3, at [https://www.ncbi.nlm.nih.gov/geo/query/acc.cgi?acc=GSE148409] and at [https://doi.org/10.17632/ybcvvmtcn9.3].

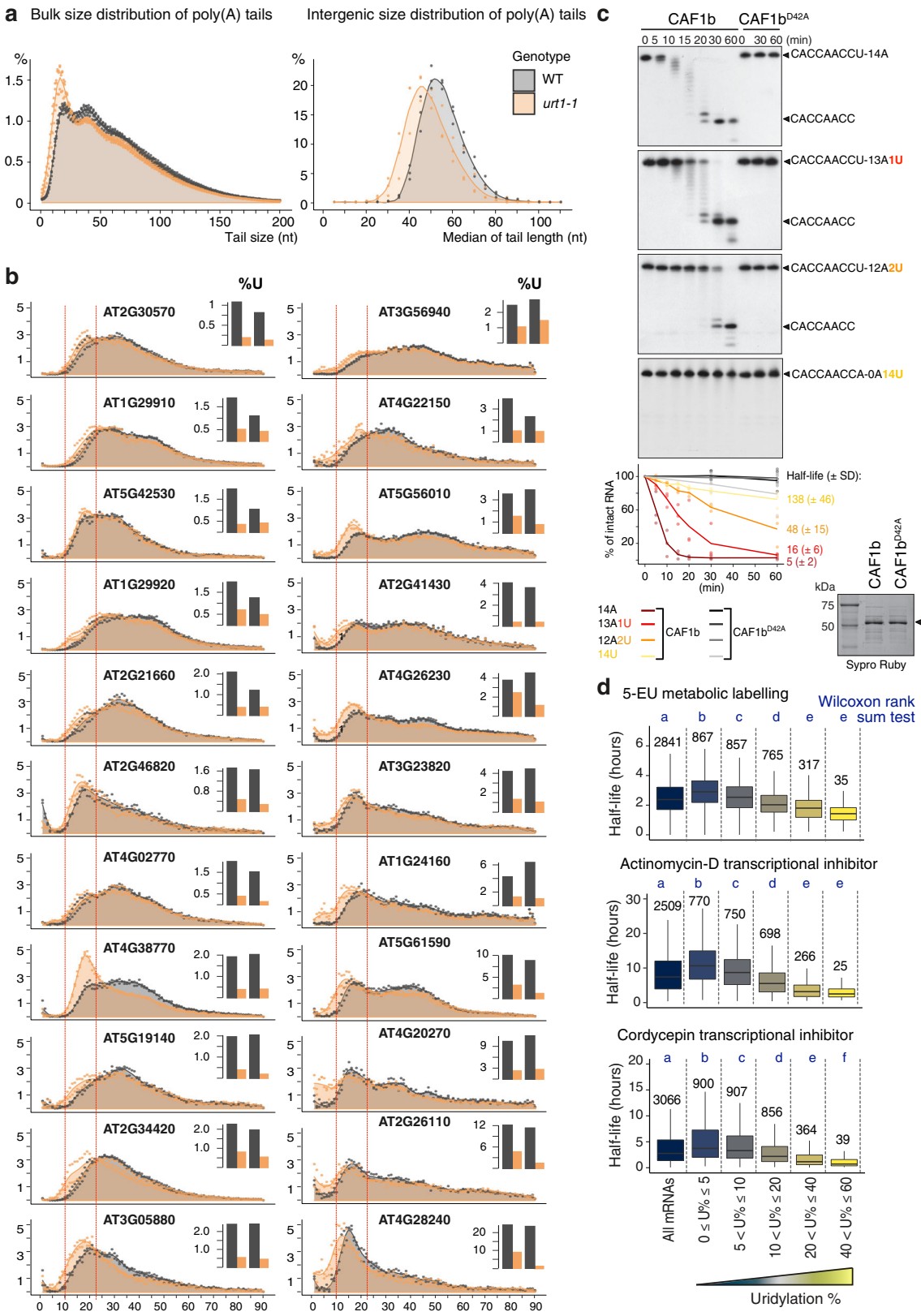

**a** Bulk size distribution of poly(A) tails Intergenic size distribution of poly(A) tails

**b** %U

**c** CAF1b CAF1b^D42A

**d** 5-EU metabolic labelling

Actinomycin-D transcriptional inhibitor

Cordycepin transcriptional inhibitor

first and last genes shown in Fig. 5b, respectively). In agreement with our previous studies[11,12], mRNA uridylation levels drop in *urt1* mutants (bar plots in Fig. 5b), and uridylation tags mostly oligoadenylated mRNAs, i.e., mRNAs with tails of <25As

(Supplementary Fig. 5). Yet, longer poly(A) tails can also get uridylated, albeit to low levels (Supplementary Fig. 5). In line with the DRS data, the profiles for poly(A) tails are modified upon loss of URT1: a slight shift toward smaller poly(A) tails is frequently

**Fig. 5 URT1-mediated uridylation shapes poly(A) tails in Arabidopsis. a** Global mRNA poly(A) profiles determined by nanopore DRS for WT and *urt1-1* plants. Individual points are shown for three biological replicates for WT (gray) and *urt1-1* (orange) and the respective average is indicated as a colored area, both for bulk (right) and intergenic (left) poly(A) size distribution. **b** Distribution profiles of poly(A) tail sizes determined by 3′RACE-seq for 22 mRNAs in WT (gray) and *urt1-1* (orange). The panels for the 22 mRNAs were ranked according to their uridylation levels in wild-type plants. The percentages of sequences were calculated for tails from 1 to 90 nt for two biological replicates. Tail length includes poly(A) stretches and eventual 3′ terminal uridines. Dashed lines indicate the 10 and 25 nt tail sizes. Small bar plots display the uridylation percentage in WT and *urt1-1* for each mRNA. **c** Deadenylation assay. Wild-type and mutated 6His-GST-CAF1b proteins (indicated by an arrow on the Sypro Ruby stained SDS-PAGE gel shown at the bottom of **c**) were incubated with 5′-labeled RNA substrates containing either an oligo(A) stretch and 0, 1, or 2 uridines, or 14 terminal uridines as indicated on the top right of each panel. The graph represents the disappearance of intact RNA substrates. Half-lives (±SD) are indicated for each RNA substrate in minutes (see Supplementary Fig. 5b). **d** Boxplot analysis comparing mRNA half-lives as determined by 5-EU metabolic labeling[37], transcription inhibition by actinomycin D[35] or cordycepin[36] vs. the percentage of mRNA uridylation in WT. Boxplot displays the median, first and third quartiles (lower and upper hinges), the largest value smaller and the smallest value larger than 1.5 interquartile (upper and lower whiskers). Uridylation percentages were measured by TAIL-seq. Datasets from three biological replicates were pooled. Numbers and letters above the boxplot represent the number of mRNAs for each category and significant statistical *p* values (two-tailed Wilcoxon rank-sum test), respectively. Exact *p* values are indicated in Supplementary Data 5b. The source data are available in Supplementary Data 3–5 and at [https://www.ncbi.nlm.nih.gov/geo/query/acc.cgi?acc=GSE148406].

observed in *urt1* (Fig. 5b). The changes in poly(A) tail profiles can be split in two effects. First, the *urt1* mutant accumulates oligoadenylated mRNAs with oligo(A) tails of 10–25 As for most of the analyzed mRNAs. This effect is particularly obvious for *AT4G38770* mRNAs, whose poly(A) tail profile is strongly impacted by loss of URT1, despite its apparently low uridylation frequency in wild type (Fig. 5b). Second, excessively deadenylated mRNAs (with tails of <10 As) accumulate in *urt1* mutants. Interestingly, the accumulation of mRNAs with tails <10 is more often observed for highly uridylated mRNAs (Fig. 5b). Hence, URT1 prevents the accumulation of excessively deadenylated mRNAs.

Two hypotheses could explain that poly(A) tails are shortened in *urt1* mutants (and conversely, shifted toward longer sizes upon URT1 overexpression in *N. benthamiana* leaves). First, uridylation by URT1 could impede deadenylation, thereby slowing down the production of oligoadenylated mRNAs. Second, URT1 could trigger the degradation of oligoadenylated mRNAs, thereby shifting the distribution toward longer poly(A) tails. Importantly, those possibilities are not mutually exclusive.

We first used in vitro assays to test whether uridylation could impede deadenylation. The Arabidopsis genome encodes two CCR4 and 11 CAF1 homologs. CAF1s are classified in three groups based on phylogenetic analyses[33]. CAF1 from group A (CAF1a and CAF1b) and group C (CAF1h–k) have been proposed to interact with NOT1, the scaffold protein of the CCR4-NOT complex[33]. We purified recombinant CCR4s and the CAF1 proteins from group A and C and tested their deadenylase activity in vitro. For unknown reasons, only CAF1b had a robust deadenylase activity under the various biochemical conditions tested. Therefore, CAF1b was used to test the intrinsic influence of uridylation on its deadenylation activity. A catalytic mutant CAF1b$^{D42A}$ was used as a negative control. CAF1b and CAF1b$^{D42A}$ were incubated with radiolabeled RNA substrates containing either 14 3′ terminal As, 13 As and 1 U, 12 As and 2 Us, or 14 Us. As expected, CAF1b fastly degraded the oligo(A) tail whereas it was markedly inhibited by 14 Us (Fig. 5c). Interestingly, the presence of a single 3′ terminal uridine delayed the degradation of the oligo(A) tail (Fig. 5c). Two 3′ terminal uridines even further impeded CAF1b activity (Fig. 5c). Therefore CAF1b activity is slowed down by the presence of a limited number of 3′ terminal uridines, but not fully inhibited. Whether this intrinsic feature is maintained upon plant CCR4-NOT complex assembly is unknown yet. Of note, the CAF1 activity within a fully reconstituted *S. pombe* CCR4-NOT complex is also slowed down by 3′ terminal uridines, albeit to a lesser extent than by guanosines and cytidines[34]. The fact that uridylation can intrinsically impede deadenylation explains, at least partly, why overexpression of URT1 in *N. benthamiana* leaves results in the accumulation of long poly(A) tails

(Figs. 3 and 4) and why loss of URT1 in Arabidopsis results in the accumulation of excessively deadenylated mRNAs, especially of mRNAs with a high uridylation level (Fig. 5b).

The second possibility to explain the change of poly(A) tail profiles in *urt1* mutants is that URT1-mediated uridylation triggers the degradation of its main targets, i.e., deadenylated mRNAs. This possibility is in line with the primordial role of mRNA uridylation, mostly investigated in *S. pombe* and mammals[5–7]. We have previously shown in Arabidopsis that 90% of uncapped *LOM1* mRNAs are uridylated, whereas only 1% of capped *LOM1* mRNAs have terminal uridines[11]. Those uncapped and uridylated mRNAs are unstable and were detected only in a *xrn4* mutant, in which cytosolic 5′-3′ RNA degradation is compromised. Those data already suggested a link between uridylation and decay in Arabidopsis, at least for *LOM1* mRNAs. Yet, a global relationship between uridylation and decay rates was not yet reported in plants. To this end, we first generated TAIL-seq libraries for three biological replicates of Col-0 plants to rank mRNAs according to their uridylation levels (Supplementary Data 5). This dataset was then compared with three independent datasets reporting transcriptome-wide mRNA half-lives in Arabidopsis[35–37]. Despite the poor correlation between the three datasets[37], we found for each comparison that the higher the propension of an mRNA to get uridylated, the shorter its half-life (Fig. 5d). This analysis reveals a robust link between uridylation and the propension of mRNAs to be degraded, similar to what has been reported in humans[6].

Even though a causal relationship between uridylation and mRNA decay has yet to be demonstrated in Arabidopsis and many molecular details remain to be solved to fully understand how URT1-mediated uridylation influences mRNA decay, the data presented here unequivocally show that URT1 shapes the global poly(A) tail profile in Arabidopsis and prevents the accumulation of excessively deadenylated mRNAs.

**URT1-mediated uridylation prevents the production of spurious siRNAs targeting mRNAs.** The accumulation of excessively deadenylated mRNAs in *urt1* mutants has no major effect on Arabidopsis growth and development, at least when plants are grown in standard conditions. Yet, introgressing the *urt1* mutation into an *xrn4* background, lacking the main cytosolic 5′-3′ exoribonuclease, had a detrimental impact on development (Fig. 6a, b and Supplementary Fig. 6): 6.5-week-old *urt1-1 xrn4-3* plants failed to set new leaves (Fig. 6a), and 9.5-week-old *urt1-1 xrn4-3* plants had severely impaired statures as compared to control plants or single mutants (Fig. 6a). Moreover, *urt1-1 xrn4-3* double mutants failed to develop inflorescences when grown under 12/12 (day/night) conditions (Fig. 6b).

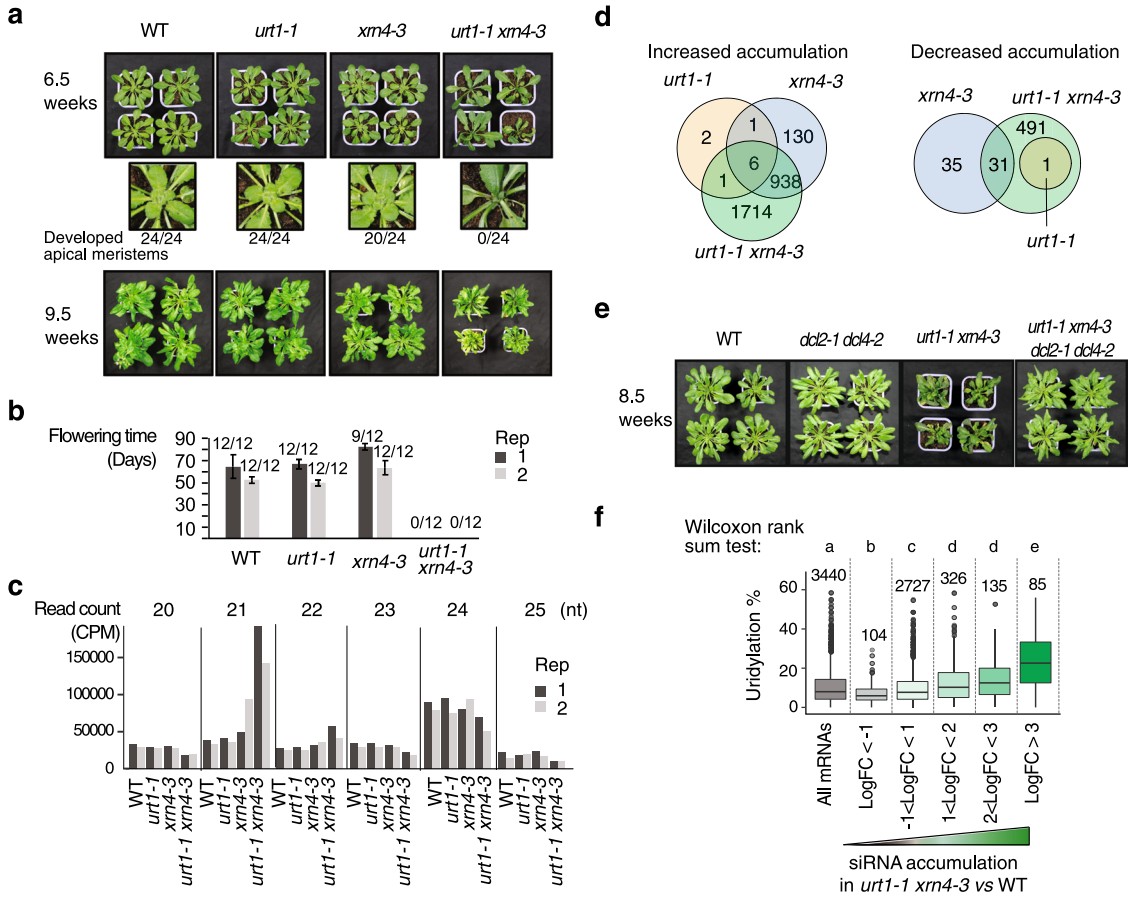

**Fig. 6 URT1-mediated uridylation prevents the production of spurious siRNAs targeting mRNAs. a** Rosettes of WT, *urt1-1*, *xrn4-3*, and *urt1-1 xrn4-3* plants. Numbers below the enlarged images of the rosette center indicate the ratio of normally developed shoot apical meristem to total numbers of plants. **b** Flowering time of WT, *urt1-1*, *xrn4-3*, and *urt1-1 xrn4-3* defined as the number of days from seed sowing until the opening of the first flower. Bar plots show the mean (±SD) of the flowering time measured for twelve plants of two biological replicates. Individual points are shown for each plants. Numbers of flowering plants/total number of plants are indicated above each bar plot. Plants in (**a**, **b**) were grown in 12-h light/12-h darkness photoperiod. **c**, **d** Small RNA-seq analyses performed for two biological replicates from 24-day-old seedlings grown in vitro in 12-h light/12-h darkness photoperiod. **c** Bars plot show the number of 21–25-nt reads as counts per million (CPM) that map to mRNAs in WT, *urt1-1*, *xrn4-3*, and *urt1-1 xrn4-3*. **d** Venn diagrams show the number of mRNAs for which siRNA levels differentially increased or decreased in *urt1-1* (orange), *xrn4-3* (blue), and *urt1-1 xrn4-3* (green) when compared to WT. **e** Rosettes of WT, *dcl2-1 dcl4-1*, *urt1-1 xrn4-3*, and *dcl2-1 dcl4-1 urt1-1 xrn4-3* plants grown in 12-h light/12-h darkness photoperiod. **f** Boxplot analysis comparing the percentage of mRNA uridylation in WT vs. the accumulation of siRNAs in *urt1-1 xrn4-3*. Boxplot displays the median, first and third quartiles (lower and upper hinges), the largest value smaller and the smallest value larger than 1.5 interquartile (upper and lower whiskers). Uridylation percentages were measured by TAIL-seq. Datasets from three biological replicates were pooled. Numbers and letters above the boxplot represent the number of mRNAs for each category and significant statistical *p* values (two-tailed Wilcoxon rank-sum test), respectively. Exact *p* values are indicated in Supplementary Data 5b. The source data are available in Supplementary Data 5 and 6, and at [https://www.ncbi.nlm.nih.gov/geo/query/acc.cgi?acc=GSE148449] and [https://doi.org/10.17632/ybcvvmtcn9.3].

The accumulation of RNA decay intermediates such as uncapped and excessively deadenylated mRNAs can be deleterious in plants because such aberrant mRNAs can erroneously trigger the biogenesis of siRNAs[38–43]. Because some illegitimate siRNAs are produced in an *xrn4* mutant[38], we checked whether *urt1-1 xrn4-3* growth and developmental defects are linked to an increased biogenesis of spurious siRNAs. We first analyzed small RNA libraries from 24-day old in vitro grown seedlings before the onset of visible phenotypes and indeed detected an increased accumulation of 21-nt siRNAs originating from mRNA loci in *urt1-1 xrn4-3* (Fig. 6c). Overall, siRNAs derived from 2659 mRNAs significantly accumulated in *urt1-1 xrn4-3* (Fig. 6d and Supplementary Data 6). Preventing siRNA production by mutating DCL2 and DCL4 in *urt1-1 xrn4-3* abrogated the growth and developmental defects associated to the *urt1-1 xrn4-3* mutation (Fig. 6e). This result demonstrates the causality between *urt1-1 xrn4-3* phenotype and the production of spurious siRNAs.

Finally, we checked whether the mRNAs that are more prone to trigger the synthesis of spurious siRNAs in *urt1-1 xrn4-3* are highly uridylated in wild-type plants. We therefore compared the TAIL-seq results to the small RNA-seq data and indeed, mRNAs that produce spurious siRNAs in *urt1-1 xrn4-3* have a significantly higher propension to uridylation in wild-type plants (Fig. 6f).

Altogether, our data reveal that URT1-mediated uridylation prevents the accumulation of excessively deadenylated mRNAs, and by doing so, avoids the production of spurious siRNAs that can target endogenous mRNAs.

## Discussion

Uridylation is now recognized as an integral step of mRNA degradation in eukaryotes. Yet, the full range of its molecular functions in assisting mRNA decay remains to be defined. Based

on the URT1 interactome and the functional analysis of URT1-mediated uridylation presented here, we propose a model integrating the dual function of URT1 in preventing excessive deadenylation and favoring the turnover of deadenylated mRNAs through the direct recruitment of decapping activators (Fig. 7). By preventing the accumulation of excessively deadenylated mRNAs, URT1-mediated uridylation protects endogenous mRNAs from triggering the synthesis of spurious siRNAs in Arabidopsis.

The composite domain organization of most TUTases is proposed to be key for the recruitment of factors that assist TUTases for the recognition of specific RNA substrates or channel the downstream molecular effects of uridylation[3,4,44,45]. However, only a few interactants of cytosolic TUTases have been identified to date. In mammals, TUT4/7 contacts the RBP Lin28, which binds Group II let-7 miRNA precursors. The presence or absence of Lin28 toggles TUT4/7 into a processive or a more distributive mode, promoting either degradation or maturation of let-7 precursors, respectively[46–48]. In Drosophila, the TUTase Tailor binds the 3′–5′ exoribonuclease Dis3l2 to form the terminal RNA uridylation-mediated processing (TRUMP) complex, which degrades a variety of structured ncRNAs in the cytoplasm[49]. In this study, we show that Arabidopsis URT1 co-purifies with several translational repressors/decapping activators, orthologs of which are known to form an intricate and dynamic interaction network in animals[13,20,22,50,51]. However, whether this network also comprises a TUTase in animals is not yet known.

A key factor at the heart of this dynamic network connecting translational repressors and decapping activators is the CCR4-NOT complex. Although not all components of CCR4-NOT were enriched in URT1 IPs, the detection of the CCR4-NOT complex scaffold subunit NOT1 alongside the other CCR4-NOT subunits CNOT10 and CNOT11 strongly supports a connection between the CCR4-NOT complex and URT1. The prime candidate for connecting URT1 to the CCR4-NOT complex is EXA1, a GYF domain containing protein, homologous to the human and fly GIGYF proteins[20,21]. GIGYF are translation repressors and decapping activators, that interact with different translation repressors via multiple interfaces, among them the GYF domain[20,21]. GIGYF interactants include the RNA helicase DDX6/Me31B, PAT1, the 5′ cap-binding protein 4EHP, and the CCR4-NOT complex[51,52]. Interestingly, orthologs of all these factors are detected in URT1 IPs, supporting the hypothesis that such an interaction network is conserved in Arabidopsis. In line with our data, a two-hybrid screen using Arabidopsis EXA1 as a bait (called GYN4 in this study) retrieved CNOT4 subunits[26], providing independent support for a physical association between EXA1 and the CCR4-NOT complex. Interestingly, the GYF domain of EXA1 recognizes a PPGF sequence[26], and such a motif is conserved in URT1 of flowering plants. A CCR4-NOT/EXA1/URT1 network may explain, at least in part, how URT1 selects its targets and why deadenylated mRNAs are preferentially uridylated. The 3′ extremity of long poly(A) tails that are either protected by PABPs or being shortened by CCR4-NOT would be poorly accessible to URT1, even though URT1 and CCR4-NOT are connected. However, once poly(A) tails get short enough to loosen their association with the last remaining PABP and when CCR4-NOT's activity is more distributive, the tethering of URT1 to CCR4-NOT via EXA1 could facilitate the uridylation of oligo(A)-tailed mRNAs.

In S. pombe and mammalian cells, uridylation is proposed to favor decapping of deadenylated mRNA by promoting the binding of the LSm1-7 complex[5,8,53], which recruits the decapping complex through a connection with Pat1[54–57]. The LSm1-7 complex preferentially binds short oligo(A) tails of <10 As[23]. Our previous TAIL-seq analyses[12] and the 3′RACE-seq data presented here show that the majority of uridylated oligo(A) tails are longer

than 10 As and comprise mostly 10–25 As. Such oligo(A) tails are likely suboptimal targets for the LSm1-7 complex. Interestingly, the direct recruitment of decapping activators by URT1 could bypass the requirement for LSm1-7 binding. We demonstrated here that the conserved M1 motif of URT1 directly binds to the decapping activator DCP5. DCP5 interacts with additional decapping factors, like DCP1 and DCP2[58]. For yet unknown reasons, neither DCP1 nor DCP2 were significantly enriched in URT1 IPs. By contrast, we detected VCS, an ortholog of human EDC4, which interacts with LSM14 via a FFD motif[29], perfectly conserved in Arabidopsis DCP5. Moreover, a LSM14-DDX6 interaction via LSM14's FDF and TFG motifs (both conserved in DCP5) is required to expose the FFD motif for EDC4 recruitment[29]. Our URT1 IP data support the idea that a DCP5/RH6,8,12/VCS connection also exists in Arabidopsis. In addition, the most straightforward interpretation of the poly(A) profiles observed upon overexpression of different URT1 versions in N. benthamiana is that the interaction of URT1's M1 motif with DCP5 promotes the degradation of oligoadenylated mRNAs (and possibly mRNAs with heteropolymeric A-rich tails). We therefore propose that the conserved M1 motif in the N-terminal IDR of URT1 contacts DCP5, which then recruits the RH6,8,12-VCS decapping activators. In addition, URT1 contacts the GIGYF-like EXA1, likely via URT1's PPGF motif. EXA1 recruits further translational repressors and decapping activators like nCBP and PAT1. Altogether these interaction networks would facilitate decapping of mRNAs with oligo(A) tails larger than those typically required for LSm1-7 recruitment.

But why has a bypass of the LSm1-7 recruitment been selected during plant evolution? A likely reason is that in plants, excessive deadenylation could trigger the biogenesis of spurious siRNAs targeting endogenous mRNAs to PTGS. Indeed, the RNA-dependent RNA polymerase RDR6, the key enzyme converting "aberrant" RNA into double stranded RNA that will be diced into siRNAs, intrinsically favors fully deadenylated mRNAs over polyadenylated mRNAs as templates[59]. We therefore propose that a key role for URT1-mediated uridylation is to avoid the accumulation of excessively deadenylated mRNAs that otherwise erroneously enter the RNA silencing pathway. In line with this hypothesis, a connection between URT1 and RNA silencing was recently suggested by the identification of URT1 as a silencing suppressor of transgenes[60]. Our results indicate two additive modes of action to explain how URT1 limits the accumulation of excessively deadenylated mRNAs (Fig. 7). First, the direct URT1-DCP5 interaction mediates a molecular connection between a TUTase and decapping activators, thereby facilitating the 5′-3′ removal of oligoadenylated mRNAs with oligo(A) tails in the 10–25 A range. Second, uridylation per se can participate in preventing excessive deadenylation by slowing down deadenylases, at least the CAF1 activity tested in this study. Finally, we have previously shown that uridylation by URT1 repairs deadenylated mRNAs to restore an extension sufficient for the binding of a PABP[12]. Although it is yet unknown how the binding of PABP to uridylated oligo(A) tails influences deadenylation or translation, binding of a PABP may also protect mRNA 3′ extremity from being accessible to RDR6. Altogether, these data illustrate the complexity of uridylation-mediated processes. Although facilitating degradation emerges as the prototypical function of mRNA uridylation, the underlying molecular mechanisms are complex and may differ across eukaryotes. This diversity is yet to be fully explored.

## Methods
**Plant material**. All Arabidopsis thaliana plants used in this study are of Columbia (Col-0) accession. T-DNA mutants were described previously: urt1-1 (Salk_087647C)[11], urt1-2 (WISCDSLOXHS208_08D)[11], xrn4-3 (SALK_014209)[61],

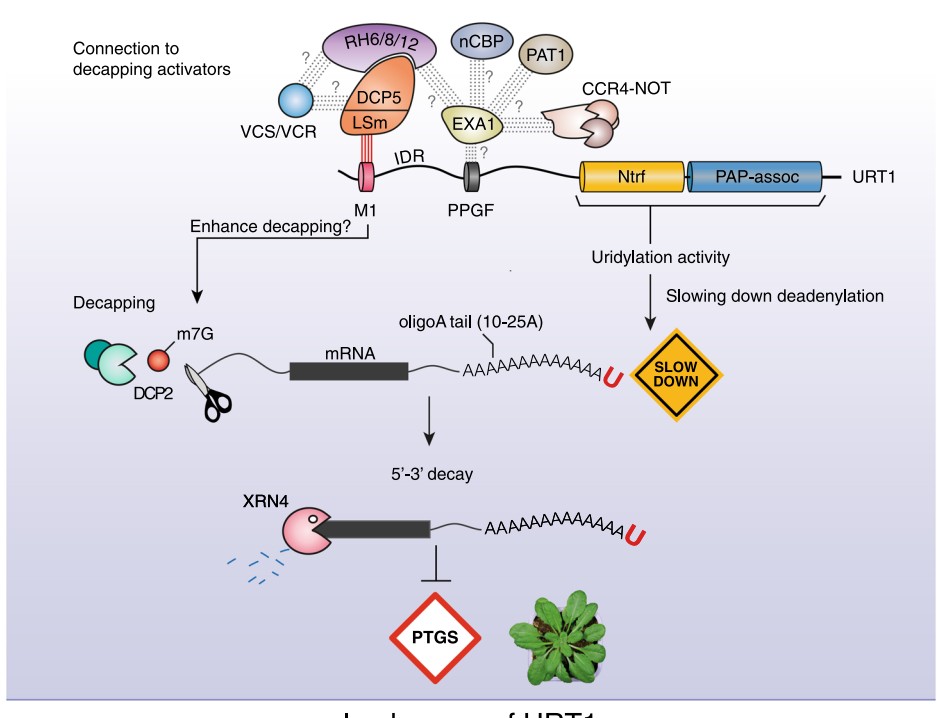

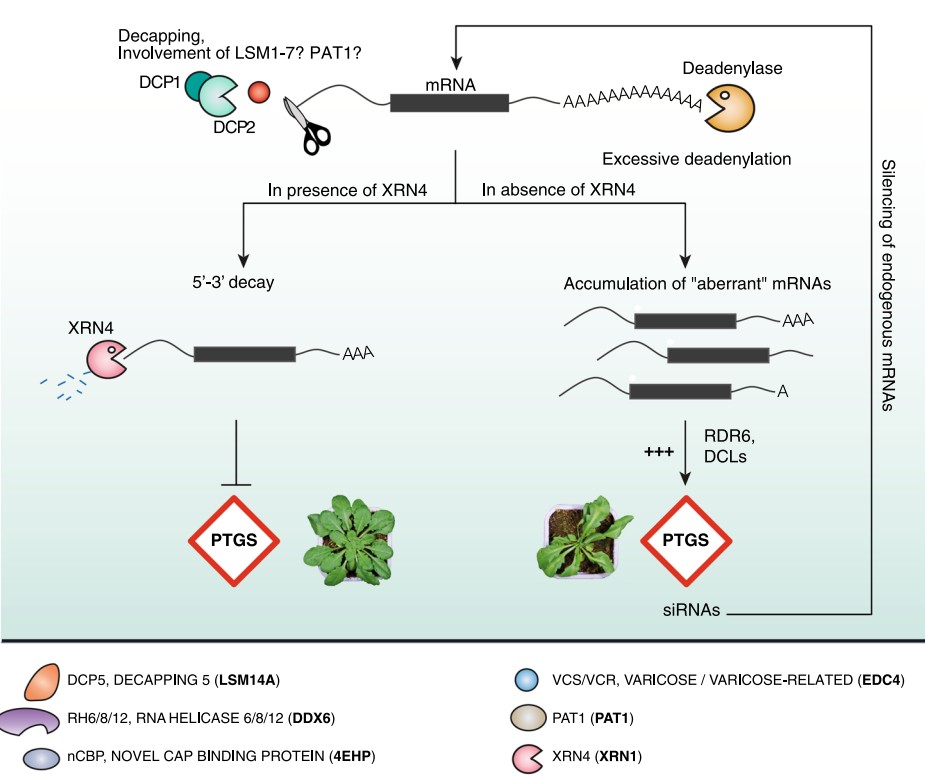

**Fig. 7 Model of URT1 mode of action.** The name correspondence of RNA decay factors conserved between Arabidopsis and humans is indicated at the bottom.

*xrn4-5* (SAIL_681E01)[62], *dcl2-1* (SALK_064627)[63], *dcl4-2* (GABI_160G05)[64], and *dcp5-1* (SALK_008881)[58]. The plant material used for RNA-seq and small RNA-seq corresponds to Arabidopsis plantlets grown for 24 days in vitro on Murashige & Skoog media with 0.8% agar and 12-h light/12-h darkness cycles (22/18 °C). For other analyses using Arabidopsis, flowers were harvested from plants grown on soil with 16-h light/8-h darkness cycles (21/18 °C). Agroinfiltration experiments were performed in leaves of *Nicotiana benthamiana* plants grown during 4 weeks on soil with 16-h light/8-h darkness cycles (22/18 °C).

**Characterization of URT1 sequence and phylogeny**. The characterization of URT1 sequences and phylogenetic analyses are detailed in Supplementary Methods. All protein sequences used for Fig. 1 and Supplementary Fig. 1 are provided in Supplementary Data 2.

**Oligonucleotides and plasmids**. Oligonucleotides and expression plasmids used for cloning are listed in Supplementary Data 7. For plant transformation, the URT1 sequence was PCR-amplified from genomic DNA and includes 5′ UTR (URT1-myc constructs) or 3′ UTR (myc-URT1 and YFP-URT1 constructs). For bacterial expression, URT1, DCP5, and CAF1b sequences were PCR-amplified from cDNA templates.

**Co-immunopurifications (IP)**. Details about samples and replicates of co-immunopurification (IP) experiments are provided in Supplementary Data 2e. For IPs without crosslinking on Arabidopsis samples, 300 mg of flower buds or seedlings were ground in 1.5 ml of ice-cold lysis buffer (50 mM Tris-HCl pH 8, 50 mM NaCl, 1% Triton X-100, protease inhibitors (cOmplete, EDTA-free Protease Inhibitor Cocktail, Roche). After cell debris removal by centrifugation (twice 10 min at 16,000 g, 4 °C), supernatants were incubated for 30 min with 50 μl of magnetic microbeads coupled to anti-c-myc antibodies or anti-GFP antibodies (Miltenyi). Beads were loaded on magnetized μMACS separation columns equilibrated with lysis buffer and washed four times with 200 μl of washing buffer (20 mM Tris-HCl pH 7.5, 0.1% Triton X-100). Samples were eluted in 100 μl of pre-warmed elution buffer (50 mM Tris-HCl pH 6.8, 50 mM DTT, 1% SDS, 1 mM EDTA, 0.005% bromophenol blue, 10% glycerol). Negative control IPs were performed under the exact same conditions with Col-0 plants.

For each IP with crosslinking step on Arabidopsis samples, 150 mg of Arabidopsis flower buds were ground during 10 min in 1.125 ml of ice-cold lysis buffer supplemented with 0.375% formaldehyde (Thermo Fisher Scientific). The crosslinking reaction was quenched by adding glycine at a final concentration of 200 mM for 5 min. After cell debris removal by centrifugation (twice 15 min at 10,000 g, 4 °C), supernatants were incubated for 45 min with 50 μl of magnetic microbeads coupled to anti-c-myc antibodies or anti-GFP antibodies (Miltenyi). Beads magnetic capture and washing steps were done according to the manufacturer's instructions, except that washes were performed with 50 mM Tris-HCl pH 7.5, 50 mM NaCl, 0.1% Triton X-100, protease inhibitors (cOmplete, EDTA-free Protease Inhibitor Cocktail, Roche). Samples were eluted in 100 μl of pre-warmed elution buffer (50 mM Tris-HCl pH 6.8, 50 mM DTT, 1% SDS, 1 mM EDTA, 0.005% bleu de bromophenol, 10% glycerol). Negative control IPs were performed with beads coupled to anti-c-myc and anti-GFP antibodies in Col-0 plants or in plants expressing the GFP alone[65]. To improve the stringency of the analysis and improve the identification of contaminant proteins, additional negative control IPs were performed using GFP-BPM6 expressing plants[66]. BPM6, encoded by AT3G43700, acts as an adaptor for Cullin3-based E3 ubiquitin ligase and has no known function related to RNA metabolism.

For each IP with crosslinking step on *N. benthamiana* samples, 500 mg of leaves were ground during 10 min in 1.5 ml of ice-cold lysis buffer supplemented with 0.375% formaldehyde (Thermo Fisher Scientific). Beads magnetic capture and washing steps were performed with the same protocol as the one described above for Arabidopsis samples. Each IP sample was performed in 2–4 biological replicates and 1–3 technical affinity replicates (Supplementary Data 2e).

**Mass spectrometry analysis and data processing**. Eluted proteins were digested with sequencing-grade trypsin (Promega) and analyzed by nanoLC-MS/MS. For IPs with crosslinking (Fig. 2a, f), digested proteins were analyzed on a QExactive+ mass spectrometer coupled to an EASY-nanoLC-1000 (Thermo Fisher Scientific). For IPs without crosslinking (Fig. 2b), digested proteins were analyzed on a TT5600 mass spectrometer (SCIEX) coupled to an Eksigent Ultra2D-plus nanoHPLC. IP data were searched against the TAIR 10 database (for *A. thaliana* samples) or the Sol Genomics Niben101 database (for *N. benthamiana* samples) with a decoy strategy. Peptides were identified with Mascot algorithm (version 2.5, Matrix Science) and data were imported into Proline 1.4 software (http://proline.profiproteomics.fr/). Proteins were validated on Mascot pretty rank equal to 1, and 1% FDR on both peptide spectrum matches (PSM score) and protein sets (Protein Set score). The total number of MS/MS fragmentation spectra was used to quantify each protein from at least four independent IPs and two independent biological replicates (see details in Supplementary Data 2e). Volcano plots display the adjusted p values and fold changes in Y- and X-axis, respectively, and show the enrichment of proteins co-purified with tagged URT1 IPs as compared to control IPs (Fig. 2a, b), the differential accumulation of proteins between URT1 and m1URT1 IPs (Fig. 2f) or the differential accumulation of proteins between m1URT1$^{D491/3A}$ and URT1$^{D491/3A}$ IPs (Supplementary Fig. 3c). The statistical analysis based on spectral counts was performed using a homemade R package that calculates fold change and p values using the quasi-likelihood negative binomial generalized log-linear model implemented in the edgeR package[67]. Common and tagwise dispersions were calculated with the implemented edgeR function by filtering out the 50% less abundant proteins that could adversely affect the dispersion estimation. The size factor used to scale samples were calculated according to the DESeq2 normalization method (i.e., median of ratios method)[68]. P value was

adjusted using the Benjamini–Hochberg method from stats R package. The gene ontology analysis for URT1 co-purifying proteins was performed using the Functional Annotation tool implemented in DAVID (v6.8)[69]. For *N. benthamiana* IPs, XIC (Extracted Ion Chromatograms)-based abundances were also calculated for peptides that map DCP5 isoforms. XIC-based quantification and statistics were performed using the Proteome Discoverer software (v2.3, Thermo Scientific) with the following parameters: Sequest and MS-Amanda algorithms with a FDR at 1%, "Top 3 Average" method, no imputation.

**In vitro pull-down assays**. Recombinant 6His-GST, 6His-GST-URT1, 6His-GST-m1URT1, 6His-GST-URT1$^{1–40}$, 6His-GST-m1URT1$^{1–40}$, 6His-MBP-DCP5, 6His-MBP-ΔLSmDCP5, and 6His-MBP-DCP5-LSM were expressed into *Escherichia coli* BL21 DE3 using plasmids listed in Supplementary 7. Conditions for protein expression and purification are detailed in Supplementary Methods. Ten pmol of each purified protein were incubated in a final volume of 500 μl of 20 mM MOPS pH 7.2, 100 mM KCl, 15% glycerol, and 0.1% Tween 20 with 100 ng/μl of RNase A for 10 min at 4 °C under rotation. Eighty microliters of glutathione sepharose resin (GE healthcare) were added to each reaction and incubated under rotation for 1 h at 4 °C. The resin was sedimented at 500 g for 5 min and washed five times with 500 μl of the same buffer without RNase A. The elution was performed by adding 60 μl of elution buffer 20 mM MOPS pH 7.2, 100 mM KCl, 15% glycerol and 0.1% Tween 20, 10 mM reduced glutathione (Sigma-Aldrich) and incubated 5 min at 4 °C before elution by centrifugation at 500 g for 5 min. Eluted proteins were separated by SDS-PAGE and stained using SYPRO Ruby dye (Bio-Rad). An Attan DIGE imager (Amersham Biosciences) or Fusion FX camera system (Vilber) was used for visualization.

**In vitro activity assays**. Recombinant 6His-GST-CAF1b and 6His-GST-CAF1b$^{D42A}$ proteins were expressed in *E. coli* BL21 using plasmids listed in Supplementary 7. Conditions for protein expression and purification are detailed in Supplementary Methods. The deadenylation test was performed in biological triplicates in 20 mM MOPS pH 7.2, 5 mM MgCl$_2$, 50 mM KCl, 7% glycerol, and 0.1% Tween 20. The oligoribonucleotides CACCAACCACU-A$_{14}$, CACCAACCACU-A$_{13}$U$_1$, CACCAACCACU-A$_{12}$U$_2$, and CACCAACCACA-U$_{14}$ were used as substrates. Purified 6His-GST-CAF1b and 6His-GST-CAF1b$^{D42A}$ proteins (30 nM) were incubated with 17.5 nM of radiolabelled substrate for 1 h at 25 °C. Aliquots were taken at different time points and separated on a 17% polyacrylamide/7 M urea gel before autoradiography. The amount of intact substrate RNA was estimated as detailed in Supplementary Methods.

**Agroinfiltration experiments in *N. benthamiana***. *Agrobacterium tumefaciens* GV3101 (pMP90) were transformed with plant expression plasmids listed in Supplementary 7 and inoculated in 10 ml of LB for 20 h at 28 °C. Pre-cultures were then centrifuged at 5000 g for 15 min. The pellets were resuspended at an OD600 of 1 in 5 ml of agroinfiltration buffer (10 mM MgCl$_2$ and 250 μM of 3′,5′-Dimethoxy-4′-hydroxyacetophenone (Sigma-Aldrich)). The cell suspensions containing the P19, URT1, and GFP constructs were mixed to a 1:1:1 ratio and infiltrated into *N. benthamiana* leaves using needleless syringes. The plant material was harvested 4 days after infiltration for RNA and protein extraction. Pictures of the infiltrated leaves were taken under UV illumination at 365 nm using a UVP Blak-Ray B-100Y UV lamp (Thermo Fisher Scientific) to detect the expression of the GFP reporter. The intensity of the GFP fluorescence was quantified using ImageJ (see details in Supplementary Methods).

**Western blot analysis**. *N. benthamiana* infiltrated leaf patches (four different leaves per sample) were ground in SDS-urea extraction buffer (62.5 mM Tris pH 6.8, 4 M urea, 3% SDS, 10% glycerol, 0.01% bromophenol blue). The samples were separated by SDS-PAGE and electrotransferred to a 0.45 μm Immobilon-P PVDF membrane (Millipore). The membrane was incubated overnight at 4 °C with DCP5 antibodies (provided by Rémy Merret and Cécile Bousquet-Antonelli, used at 1/10,000 dilution), URT1 antibodies (described in ref. [12] and used at 1/10,000 dilution), MBP antibodies (Invitrogen, used at 1/5000 dilution), monoclonal c-myc antibody (Roche, used at 1/10,000 dilution). Polyclonal and monoclonal antibodies were detected by goat anti-rabbit or anti-mouse IgG coupled to peroxidase (Invitrogen, used at 1/10,000 dilution), respectively, and using Lumi-Light Western Blotting Substrate (Roche). Pictures were taken with a Fusion FX camera system. The PVDF membranes were stained with 0.1% Coomassie Brilliant Blue R-250, 7% acetic acid, 50% methanol) to monitor loading.

**Northern blot analysis**. For each sample, four infiltrated patches pooled from different leaves were harvested from *N. benthamiana* four days after agroinfiltration. Total RNA was extracted using Tri-Reagent (Molecular Research Center), followed by extraction with acid phenol:chloroform:isoamyl alcohol and RNA precipitation with ethanol. Five micrograms of RNAs were then separated on a 1.5% agarose gel containing 0.2-M MOPS pH 7.0, 20 mM sodium acetate, 10 mM EDTA, 5.55% formaldehyde and transferred onto a nylon membrane (Amersham's Hybond N+, GE Healthcare). After transfer, RNAs were UV crosslinked at 120 mJ/cm² for 30 s with a Stratagene Stratalinker. The membrane was stained with methylene blue and a picture was taken for illustrating equal loading between

samples. After destaining, membranes were incubated with PerfectHyb hybridization buffer (Sigma-Aldrich) for 30 min at 65 °C and hybridized overnight at 65 °C with radiolabeled probes to detect *PR2* (Niben101Scf04869g03002.1) or *GFP* mRNAs. *PR2* or *GFP* PCR amplicons of about 500 pb (Supplementary Data 7) were used as templates to produce random labeled probes using the Decalabel DNA Labelling kit (Thermo Fisher Scientific) and [α³²P]-dCTP according to manufacturer's instructions. Radiolabeled probes were purified on a Sephadex G-50 matrix before hybridization. The membranes were exposed to a photosensitive Phosphor screen and visualized with an Amersham Typhoon IP Biomolecular Imager (GE Healthcare Life Sciences).

**3′ RACE-seq library preparation and data processing.** Total RNA was extracted using Tri-Reagent (Molecular Research Center) from *N. benthamiana* infiltrated leaves or Arabidopsis flowers. Ten pmoles of a 5′-riboadenylated DNA oligonucleotide (3′-Adap RACEseq, Supplementary Data 7) were ligated to 5 μg of total RNA using 10 U of T4 ssRNA Ligase 1 (NEB) in a final volume of 50 μl for 1 h at 37 °C and 1X T4 of RNA Ligase Reaction Buffer (NEB, 50 mM Tris-HCl pH 7.5, 10 mM MgCl2, 1 mM DTT). The ligation products were purified from reagents and non-ligated adapter molecules with Nucleospin RNA Clean-up columns (Macherey Nagel). cDNA synthesis was performed in 20 μl reaction that contains 2–3 μg of purified ligated RNA, 50 pmol of the 3′-RT oligonucleotide (Supplementary Data 7), 10 nmol of dNTP, 0.1 μmol of DTT, 40 U of RNaseOUT (Invitrogen), 200 U of SuperScript IV reverse transcriptase (Invitrogen), and 1X of SuperScript IV RT buffer (Invitrogen). Reactions were incubated at 50 °C for 10 min, and then at 80 °C for 10 min to inactivate the reverse transcriptase. Two nested PCR amplification rounds of 30 and 20–30 cycles, respectively, were then performed. PCR1 was run using 0.5–2 μl of cDNA, 10 pmol of gene-specific primer (Supplementary Data 7), 10 pmol of RACEseq_rev1 primer (Supplementary Data 7), 10 nmol of dNTP, 1 U of GoTaq DNA Polymerase (Promega), and 1X of Green GoTaq Reaction Buffer (Promega) in a 20-μl final volume. The conditions for PCR1 were as follows: a step at 94 °C for 30 s; 30 cycles at 94 °C for 20 s, 50 °C for 20 s, and 72 °C for 30 s; a final step at 72 °C for 30 s. PCR2 was performed using 1 μl of PCR1 product, 10 pmol of gene-specific primer (Supplementary Data 7) and 10 pmol of a TruSeq RNA PCR index (RPI, Supplementary Data 7), 10 nmol of dNTP, 1 U of GoTaq DNA Polymerase (Promega), and 1X of Green GoTaq Reaction Buffer (Promega) in a 20 μl final volume. The conditions for PCR2 were as follows: a step at 94 °C for 1 min; 20–30 cycles at 94 °C for 30 s, 56 °C for 20 s, and 72 °C for 30 s; a final step at 72 °C for 30 s. All PCR2 products were purified using one volume of AMPure XP beads (Agencourt). Library was paired-end sequenced with MiSeq (v3 chemistry) with 41 × 111-bp cycle settings. After initial data processing by the MiSeq Control Software v 2.6 (Illumina), base calls were extracted and further analyzed by a set of homemade scripts detailed in Supplementary Methods. In addition to the analysis based on the Illumina base-calling software, the poly(A) sizes shown in Supplementary Fig. 2b were also estimated using the TAILseeker software[9] (v3.1, https://github.com/hyeshik/tailseeker) as detailed in Supplementary Methods. Similar differences in poly(A) tail size distributions were observed using standard Illumina base-calling software and the Tailseeker algorithm, designed to limit poly(A) tail length overestimation (compare Fig. 3g and Supplementary Fig. 3b). Distribution profiles shown in Figs. 3–5 and in Supplementary Fig. 3–5 display the percentages of sequences according to poly(A) tail sizes calculated for tails from 1 to 90 nucleotides. The percentages were calculated using the total number of sequences with tails from 1 to 90 nucleotides, with the exception of Supplementary Fig. 5a where the percentages were calculated using the total number of uridylated sequences. Tail length takes into account the number of As and potential 3′ added nucleotides. For *GFP* and *NbPR2* mRNAs (Figs. 3, 4 and Supplementary Figs. 3, 4), poly(A) distributions were generated by considering only 3′ extremities that map in a region of ±50 nt around the main polyadenylation site. The numbers of sequences obtained at each processing step of 3′RACE-seq libraries are provided in Supplementary Data 3.

**Nanopore DRS and data processing.** Total RNA was extracted from wild-type and *urt1-1* flowers using Tri-Reagent (Molecular Research Center) following the manufacturer's recommendations. Capped mRNAs were enriched using GST-eIF4E-K119A protein[70]. Nanopore direct-RNA libraries were prepared from 5 μg of Arabidopsis cap-enriched mRNA with 150 ng of yeast poly(A)-enriched RNA using Direct RNA Sequencing Kit (ONT, SQK-RNA002)[71]. Sequencing was performed with MinION device (Flow cell type R9.4.1; RevC). Sequencing reads were basecalled using Guppy 4.0.11 (Oxford Nanopore Technologies) and mapped to the TAIR10 reference transcriptome (Ensembl release 45, merged cDNA and ncRNA) using MiniMap 2.17. Poly(A) lengths were estimated using Nanopolish 0.13.2. The numbers of sequences obtained for each nanopore library are provided in Supplementary Data 4. Bulk distribution profiles shown in Fig. 5a display the percentages of sequences according to poly(A) tail sizes calculated for all sequences. Intergenic distribution profiles shown in Fig. 5a display the median distribution of poly(A) tail sizes for mRNAs detected with at least 100 reads in each replicate and genotype (2644 transcripts).

**TAIL-seq library preparation and data processing.** TAIL-seq libraries were generated from three biological replicates of Arabidopsis Col-0 flower buds. Total RNA was extracted using Tri-Reagent (Sigma-Aldrich), treated with DNase I

(Thermo Fisher Scientific), and purified using the RNeasy MinElute Clean-up (Qiagen). Per sample, three individual ribodepletion on 10 μg of RNA each in a final volume of 10 μl were performed using the RiboMinus Plant kit (Thermo Fisher scientific) following the manufacturer's instructions. Ribodepleted RNA were then ligated to 10 pmol of a biotinylated 3′ adapter, (3′-Adap TAIL-seq, Supplementary Data 7) using 10 units of T4 RNA ligase 1 (NEB) in a final volume of 10 μl for 1 h at 37 °C. RNAs were partially digested with 0.001 unit of RNase T1 (Invitrogen) in a final volume of 80 μl for 5 min at 50 °C. RNAs were then purified with streptavidin beads (Dynabeads M-280 Streptavidin), phosphorylated using T4 PNK (NEB), and gel purified on a denaturing 6% polyacrylamide gel (Novex, 300–1200nt). Purified RNAs were ligated to 5 pmol of 5′ adapter (5′-Adap TAIL-seq, Supplementary Data 7), using 8 units of T4 RNA ligase 1 (NEB) and 8 nmol of ATP in a final volume of 8 μl for 1 h at 37 °C. cDNAs were synthesized using Superscript III (Invitrogen) and 50 pmol of RT primer (3′-RT, Supplementary Data 7). Finally, cDNAs were amplified using the DNA Phusion Polymerase master mix (Thermo Fisher Scientific) with 25 pmol of TAIL-seq-fw primer (Supplementary Data 7) and 25 pmol of a TruSeq RPI (Supplementary Data 7) in a final volume of 50 μl. PCR conditions were as follows: a step at 98 °C for 30 s; 19 cycles at 98 °C for 10 s, 60 °C for 30 s and 72 °C for 45 s; a final step at 72 °C for 5 min. Libraries were first purified on 6% polyacrylamide gel (Novex) to extract DNA fragment from 300 to 1000 nucleotides and further purified using one volume of AMPure XP beads (Agencourt). Library concentrations were determined using a Qubit fluorometer (Invitrogen). Quality and size distribution were assessed using a 2100 Bioanalyzer system (Agilent). Library were paired-end sequenced with MiSeq (v3 chemistry) with 41 × 111-bp cycle settings. The base calling-based pipeline was adapted from ref. [12] and is detailed in Supplementary Methods. Uridylation percentages were calculated for 3340 mRNAs detected with at least 20 reads using the pooled dataset from the three biological replicates. The number of sequences obtained at each processing step of TAIL-seq libraries and the calculated uridylation percentages are provided in Supplementary Data 5.

**Small RNA-seq library preparation and data processing.** Total RNA was extracted using Tri-Reagent (Molecular Research Center) from WT, *urt1-1*, *xrn4-3*, and *urt1-1 xrn4-3* 24-day-old seedlings, two biological replicates each, and subsequently treated with DNase I (Thermo Fisher Scientific). Small RNA libraries were prepared and sequenced at Fasteris (http://www.fasteris.com). Libraries were generated from 3 μg of DNase-treated RNA using the Illumina TruSeq Small RNA protocol after size selection of 18–30-nt RNA fragments on a denaturing polyacrylamide gel. Libraries were sequenced on a HiSeq 2500 (HiSeq high-output mode, 1 × 50 bp). The base calls were acquired from HiSeq 2500 after processing by Illumina RTA 1.18.61.0 and CASAVA pipeline v.1.8.2. Details about further data processing are provided in Supplementary Methods. The numbers of sequences obtained for small RNA-seq libraries and the list of mRNA loci that show differential small RNA accumulation are provided in Supplementary Data 6.

**Statistics and reproducibility.** Statistical analyses were performed using R 3.6.1, Rstudio 1.2, and the following R packages: edgeR 3.26.5, stats 3.6.1, multcompView 0.1-8. For all analyses, a *p* value of 0.05 was defined as the threshold of significance. For each figure, the exact value of *n* and the test used for the statistical analysis are indicated in the figure or in the corresponding legend. Fold change and *p* values in Fig. 2a, b, f and Supplementary Fig. 3c were computed using the quasi-likelihood negative binomial generalized log-linear model implemented in the edgeR package[67]. Statistical significance shown in Fig. 3 and Supplementary Figs. 3b, 4b was obtained using Pairwise Wilcoxon Rank-Sum tests with data considered as paired (two-tailed). Statistical significance shown in Figs. 5d and 6f was obtained using Pairwise Wilcoxon Rank-Sum tests (two-tailed) with data considered as unpaired. Statistical significance shown Supplementary Fig. 3d was obtained using two-tailed t-test (Proteome Discover software v2.3, Thermo Scientific). For box plot analyses in Figs. 5 and 6, the upper and lower edges correspond to the first and third quartiles, respectively. The median is indicated by a horizontal bar and whiskers show data range except far-outliers. Differential statistical analysis shown in Fig. 6d was performed using the edgeR package and its implemented negative binomial generalized log-linear model. Half-lives shown in Fig. 5c and Supplementary Fig. 5b were calculated using a quasi-Poisson regression model of the R stats package. All *p* values were adjusted using the Benjamini–Hochberg method.

For each graph, the number of independent replicates is indicated directly on the figures. For Figs. 2d, 3b, 3d, 4b, 5c and Supplementary Figs. 2a–c, 4a, representative images of 2, 2, 2, 3, 3, 3, 4, 2, and 2 independent experiments, respectively, are shown.

**Reporting summary.** Further information on research design is available in the Nature Research Reporting Summary linked to this article.

## Data availability

NGS datasets generated during this study have been deposited in NCBI's Gene Expression Omnibus[72] and are accessible through GEO Series accession number GSE148449. GEO Series accession numbers for individual datasets are GSE148406 for 3′ RACE-seq in Arabidopsis, GSE148409 for 3′RACE-seq in *N. benthamiana*, GSE148417 for TAIL-seq, and GSE148427 for small-RNA seq.

Mass spectrometry proteomics raw data have been deposited to the ProteomeXchange Consortium via the PRIDE partner repository[73] with dataset identifiers PXD018672 and PXD022676 for Arabidopsis and *N. benthamiana*, respectively.

Raw data for Nanopore DRS have been deposited at ENA with the accession number PRJEB40438.

Source data for all figures in the paper, including raw data underlying graphs and uncropped versions of gels or blots presented in the figures, are available as Mendeley data: https://doi.org/10.17632/ybcvvmtcn9.3.

The raw intensity files (.cif files) used to test the TAILseeker3 software (results shown in Supplementary Fig. 3) have not been deposited in a public repository because of their large size but are available from the corresponding author on request.

Web links for associated raw data are indicated in each figure legends. All biological materials used in this study, plant lines, and plasmids are available from the authors or the indicated sources.

## Code availability
Bioinformatic pipelines including python and bash source code for 3′RACE-seq and TAIL-seq analyses are available as Mendeley data: https://doi.org/10.17632/v8d9bd692c.1.

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

## Acknowledgements

The authors gratefully acknowledge Nicolas Baumberger for helping with DCP5 purification, Heike Lange and Damien Garcia for critical reading of the manuscript, Timothée Vincent and IBMP bioinformatics core facility and IT service for helping with the implementation of TAILseeker tools and providing computing resources. We also thank Rémy Merret and Cécile Bousquet-Antonelli (Perpignan, France) for providing anti-DCP5 antibodies. This research was funded by the Centre National de la Recherche Scientifique (CNRS), by an IdEx grant from the University of Strasbourg (H.Z.) and by a research grant from the French National Research Agency ANR-15-CE12-0008 (D.G.). All mass spectrometry analyses, the purchase of the mass spectrometer TripleTOF 5600 (SCIEX), and salaries for E.F. and C.d.A. were supported by a funding from the state managed by the French National Research Agency as part of the "Investments for the Future" program under the framework of the LABEX: ANR-10-LABX-0036_NETRNA and ANR-17-EURE-0023. The authors also acknowledge the funding of a QExactive Plus mass spectrometer (ThermoFisher) by an IdEx grant from the University of Strasbourg. The funders had no role in study design, data collection and analysis, decision to publish, or preparation of the manuscript.

## Author contributions

Conceptualization: D.G. and H.Z.; methodology: E.F., F.M.S., C.P., and P.H.; software: H.Z. and D.P.; formal analysis: H.Z., D.P., P.K., J.C., and L.K.; investigation: H.S., C.d.A., E.F., Q.S., L.P., S.K., S.M., C.P., P.H., and H.Z.; data curation: H.Z. and L.K.; writing—original draft: D.G. and H.Z.; writing—review & editing: H.S. C.d.A., and A.D.; visualization: H.S., C.d.A., and H.Z.; supervision: D.G., H.Z., and P.H.; funding acquisition: D.G. and H.Z.

## Competing interests

The authors declare no competing interests.
