## [Peer Review File · Nature Communications]

REVIEWER COMMENTS

Reviewer #1 (Remarks to the Author):

“Molecular connection between the TUTase URT1 and decapping activators” by Scheer et al. Scheer et al present a bioinformatic and experimental analysis of *Arabidopsis thaliana* URT1, its interacting proteins and RNA substrates. First, they analyze the aa composition of 87 plant URT1 homologs and identified and mapped 3 motifs in the disordered N-terminal domain of mostly land plants. The bioinformatic analysis is well described and carried out. They proceeded to identify URT1 interacting proteins by IP with and without formaldehyde cross linking, revealing URT1 as part of a larger complex/network. By mutational analysis they identified the M1 motif as the motif interacting with DCP5. This analysis is well carried out and leaves little doubt that the interaction is happening *in vivo*.

Next, the authors characterize the effect of Urt1 and mutants on the expression of GFP reporter. Inactive Urt1 leads to a slight shift in the GFP reporter mRNA. They then analyzed the GFP mRNA 3' tails by cRACE, and found that ectopically expressed URT1 uridylates mRNA tails with longer A-tails when endogenous URT1, and expression of a catalytically dead Urt1 shifts the A-tail lengths towards the accumulation of shorter A tails. Finally, spurious siRNA generation was analyzed and a role for Urt 1 in preventing spurious siRNA generation was identified.

The conclusions are novel, and characterize a largely unknown interactome of Urt1. This is a very well thought through and executed study, and I only have minor comments:

1. Please define ALL enzyme names at first mention (e.g. CCR4-Not (Carbon Catabolite Repression—Negative On TATA-less))
2. While not required, it would be interesting to know if the motifs M1, M2 and PPGF motifs are also found in other Eukaryotic TUTases, or are specific to plants. Maybe the authors can comment on this? Also, could the authors speculate why the M1, M2 and PPGF motifs are only found in a subset of plant sequences? Do other plants not require these interactions?
3. Fig 2, with 8 replicates I would like to see error bars in a and b. Is Figure 2d an SDS gel? Please clarify.
4. Figure 3, it is unclear why figures f and g are presented in 2 panels (one for rep 1-3, and one for 4-6, each). An average with error bars would be much more informative here. Same for Figure 4e, please change to mean with error bars, and figure 5. Or, for Figure 5, show one representative and move second to supplement. The figure is a bit confusing with that many bar graphs and having the rep2 moved to supplements should help simplify.
5. Figure 4a, what is the half life of the RNA? While not necessary, it would be nice to have a number (error bars?) in addition to the gel.

Reviewer #2 (Remarks to the Author):

In this paper, the authors provide a thorough analysis of the factors that interact with the plant URT1 uridyl transferase, and together with studies using mutant URT1 genes, describe a likely pathway by which Urt1 affects the half-life and pathway of degradation of the most polyadenylated mRNAs in plants. The authors have previously shown that URT1 uridylates the oligoA tails (typically <25 nts), adding a small number of uridines, to the oligo(A) tails remaining after deadenylation. In many cases this is sufficient to restore a PABP binding site which could then stabilize those oligoadenylated mRNAs. Here they provide a much more detailed analysis of the overall role of URT1 in plant mRNA metabolism.

The most physiologically relevant data is the effect of URT1 expression on the polyA tails of 22 different *Arabidopsis* mRNAs which was done by comparing the null URT1 mutant with wild-type

cells. These results confirm that a major role for uridylation by URT-1 is to allow accumulation of the "oligoadenylated" RNAs which are often uridylated, and show that in the absence of uridylation there are many fewer of these "oligoadenylated" RNAs. Note that these data conflict with the idea that uridylation interacts with the decapping machinery to degrade RNA, although it is consistent with the idea that it acts to stabilize the oligoadenylated mRNAs, and possibly by recruiting these factors which inhibit translation, such as dhh1, prevent its translation. The data is consistent with URT-1 interacting with the deadenylation complex; however this interaction primarily may result in uridylating and stabilizing oligoadenylated RNA, not promoting their degradation. The data also likely provide an explanation for why the GFP-reporter experiments resulted in constant mRNA but reduced protein.

The biological relevance of this pathway to preventing siRNA production which is detrimental to the organism is nicely shown and provides a rationale for the existence of this pathway in plants, and should also be indicated in their final figure. They might want to change the title to reflect these results.

Specific comments:

1. URT1 has a catalytic domain near the C-terminus and the N-terminal half of the protein is intrinsically disordered. They identify three small conserved motifs within this disordered region, that are conserved in a wide variety of plant species. It would be helpful to indicate the sequences of these motifs in Fig. 1 rather than in the supplemental figures, since it wasn't immediately clear that M1 and M2 are very different in sequence..
2. Using proteomics approaches they identify factors that interact with UTR1 (not necessarily direct binding partners, but potential complexes UTR1 interacts with), using crosslinked samples. Here they find the CCR/NOT deadenylation complex as well as a number of known translational repressors and one decapping activator, DCP5 (Lsm14). Using the same approach without crosslinking provide a much simpler pattern, with DCP5, eIF4G and decapping activators including 3 ddx6 type RNA helicases..
3. They demonstrate that one of these proteins, DCP5, directly binds URT1 and that the M1 element in URT1 is required for binding, as is the LSm domain of DCP5. It would be helpful to comment on whether these two domains are sufficient for the interaction. Does mutating M2 have any effect?
4. The rest of the paper is focused on changes that occur on the polyA tail of a GFP reporter mRNAs that result from expression of URT1 and various URT1 mutants in the presence of the endogenous tobacco uridyl transferases. Expression of active UTR1 reduces GFP protein expression, while catalytically inactive URT1 had no effect. However expression of active URT1 had no effect on GFP mRNA levels. Active URT1 expression resulted in increased uridylation, with uridylation of long polyA tails as well as "oligoadenylated" tails. Overall there was an increased amount of long polyA tails. A similar effect was seen on an endogenous mRNA, PR2 mRNA. This non-physiological result (presumably requires overexpression of uridyltransferases), uncovered an effect of uridylation on long polyA tails which resulted in the decreased deadenylation, likely explained by the biochemical properties of the CCR4 enzyme.
5. The role of the M1 region, which interacts with the dcp5 factor, could be to recruit the factors, that prevent translation of the mRNA (ddx6 [dhh1], pat1) and actually help stabilize the oligoadenylated mRNA. This would be similar to the accumulation of deadenylated mRNA after stress in yeast, shown by Collier and Parker, providing a pool of mRNA which could then be readenylated for future translation.
6. I am unclear about the origin of the "A-rich tails". It seems unlikely that they come from inaccurate polyadenylation. More likely they may come from incomplete deadenylation, followed by incorporation of a non-templated nucleotide and then cytoplasmic adenylation.
7. The final model figure needs to show more clearly that a major role of uridylation is to prevent the unregulated degradation of oligoadenylated RNA, which can result in siRNA production. The tight coupling of uridylation to deadenylation by interacting with the NOT1/CCR4 complex may primarily serve to do this, and not to direct complete degradation of the mRNA.

Reviewer #3 (Remarks to the Author):

In this manuscript, Scheer et al, report the connection between the TUTase URT1 and the decapping activators. The authors found that URT1 contain a conserved N-terminal intrinsically disordered region (IDR) and identified two short linear motifs (SLiMs) . Then the authors identified potential interaction proteins of URT1 through mass-spectrometry analyses of purified URT1 complex. The authors showed that SLiM1 mediates the interaction between DCP5 and URT1. The authors further show transient expression of URT1 can increase the ratio long poly-A containing GFP mRNAs gene in *Nicotiana benthamiana* leaves. The authors showed 3' terminal 1 or 2 Us can prevent the degradation of poly-A tails by CAF1b in vitro. The authors propose that ectopic expression of URT1 may trigger the degradation of oligoadenylated GFP mRNAs or prevent deadenylation of mRNAs, and thereby change the poly A profiles. The authors show that transient expression of an inactive UTR1D491/3A increased the accumulation oligoadenylated (16-25 nt A) GFP mRNAs with U-tails, which depends on the SLiM1 domain. The authors suggest that UTR1D491/3A may have a dominant negative impact on endogenous URT1 of *Nicotiana benthamiana* by depleting factors degrading uridylylated oligoadenylated mRNA through its SLiM1 domain. The authors further show that in *Arabidopsis* *urt1* mutant, a slight shift toward to short poly A tails occurs. The authors further show that absence of URT1 and XRN4 can trigger the production of siRNAs.

Concerns are:

The manuscript describes several different aspects of URT1. Each of them seems to be incomplete story and need additional evidences to support the conclusion.

The claim that URT1 may trigger the degradation of oligoadenylated GFP mRNAs or prevent deadenylation of mRNAs need support from experimental evidences besides speculations from sequencing data.

The majority of experiments were transient overexpression experiments, which may not reflect the normal physiological condition.

Is IDR also conserved in other organisms or specific to plants

Page 3, "Poales, which include key cereal crops, are among the plants whose genome encodes two URT1 paralogs, URT1A and URT1B" Paralog means homologs with different functions. Are the authors sure that URT1A and URT1b have different functions with URT1?

All the MASS experiments were performed using overexpression lines. The authors should confirm the interaction under native conditions using native promoters; at least show URT1 indeed interacts with DCP5 under native condition (native promoters).

Page 5, Please provide explanation of the shift towards longer poly(A) tails without U-tail induced by URT1 expression. Two possibilities provided here cannot explain this observation.

Figure 3, what happens to the GFP uridylation when URT1P618L is expressed. This one should give additional confirmation for the result since it has a weak activity and may have intermediate impact on GFP uridylation.

The authors examined the impact of 1 or 2 U on CAPF1. What is impact of long u-tail on CAPF1 activity? Can URT1 add long U tails?

Figure 3 and 4, How was the size distribution of u-tailed or non-u-tailed RNAs calculated? Using

total readings or using total U-tailed RNA or non. U-tailed RNAs readings to calculate the ratios of various sizes? It seems the total readings were used. If so, the observations may be due to the ratio changes of uridylated populations. It will nice to have a size distribution in u-tailed or non-u tail RNA populations

Figure 3e, the ratios of U-tailed GFP mRNAs in different UTR1D491/3A replicate (1, 2,3; vs 4, 5, 6) varies. Any explanation on this?

The claim UTR1D491/3A triggers mRNA degradation is weak, as no evidence to show u-tailed GFP and PR2 mRNAs decay faster than controls. Similar amount of GFP mRNAs were observed (Figure 3). Moreover, there is no evidence to show that URT1 depletes degradation components. The increased PR2 levels could be caused by different inducing levels. mRNA decay experiment is needed to examine the effect of URT1 on mRNA stability.

We thank the reviewers for their constructive comments. We have answered each comment and present new data that further support our conclusions. The main points of the new data are:

- URT1 and DCP5 also co-purify when both proteins are expressed at physiological levels in *Arabidopsis*.
- Nanopore direct RNA sequencing demonstrates that URT1 shapes mRNA poly(A) tail profiles in *Arabidopsis*.
- The comparison of our TAIL-seq data with published datasets reveals that mRNAs with high uridylation levels have shorter half-lives.

As suggested by reviewers, we have changed the title to better reflect our findings and reorganized the result part to focus on the three novel findings reported in our study: (1) a TUTase interacts with decapping activators, (2) uridylation shapes poly(A) tails by preventing the accumulation of excessively deadenylated mRNAs and (3) URT1-mediated uridylation prevents the biogenesis of spurious siRNAs targeting mRNAs.

POINT-BY-POINT ANSWERS TO REVIEWER COMMENTS

Reviewer #1 (Remarks to the Author):

Scheer et al present a bioinformatic and experimental analysis of *Arabidopsis thaliana* URT1, its interacting proteins and RNA substrates. First, they analyze the aa composition of 87 plant URT1 homologs and identified and mapped 3 motifs in the disordered N-terminal domain f mostly land plants. The bioinformatic analysts is well described and carried out. They proceeded to identify URT1 interacting proteins by IP with and without formaldehyde cross linking, revealing URT1 as part of a larger complex/network. By mutational analysis they identified the M1 motif as the motif interacting with DCP5. This analysis is well carried out and leaves little doubt that the interaction is happening in vivo.

Next, the authors characterize the effect of Urt1 and mutants on the expression of GFP reporter. Inactive Urt1 leads to a slight shift in the GFP reporter mRNA. They then analyzed the GFP mRNA 3' tails by cRACE, and found that ectopically expressed URT1 uridylates mRNA tails with longer A-tails when endogenous URT1, and expression of a catalytically dead Urt1 shifts the A-tail lengths towards the accumulation of shorter A tails. Finally, spurious siRNA generation was analyzed and a role fir Urt 1 in preventing spurious siRNA generation was identified.

The conclusions are novel, and characterize a largely unknown interactome of Urt1. This is a very well thought through and executed study, and I only have minor comments:

1. Please define ALL enzyme names at first mention (e.g. CCR4-Not (Carbon Catabolite Repression—Negative On TATA-less))

Acronyms for proteins and genes are now defined at first mention.

2. While not required, it would be interesting to know if the motifs M1, M2 and PPGF motifs are also found in other Eukayotic Tutases, or are specific to plants. Maybe the authors can comment on this?

We do not know yet whether these three motifs are found in certain non-plant TUTases but for instance, those three SLiMs are absent from human TUT4 or TUT7. Due to their small size, SLiMs are only detected by phylogenetic analyses. Before we could compare SLiMs between plants and non-plant eukaryotic TUTases, we must resolve the evolutionary relationship of TUTases across a large number of species. We definitely agree that the possible existence of SLiMs in non-plant TUTases is an interesting information. However, we think that this question should be addressed in a different study to focus the present work on the function of URT1-mediated uridylation in plants.

Also, could the authors speculate why the M1, M2 and PPGF motifs are only found in a subset of plant sequences? Do they other plants not require these interactions?

At least one URT1 protein comprising both M1 and M2 motifs is present in all land plants. This finding indicates a strong selection pressure for both motifs while the primary sequence of the surrounding intrinsically disordered region is variable. Because M1 and M2 are conserved in all land plants, we propose that these SLiMs play key roles for URT1 function.

The Poales, which represent a large and diverse order of monocotyledons, encode two URT1 homologs that we named URT1As and URT1Bs, respectively. URT1As do possess both M1 and M2 motifs. As compared with URT1As and other plant URT1 proteins, URT1Bs have different expression patterns and different motifs in a shorter IDR. It is therefore possible that URT1As and URT1Bs have different protein interaction networks. However, we did not experimentally address this possibility yet.

The PPGF motif seems less conserved across land plants than the M1 and M2 motifs. However, if the function of the PPGF motif is to connect URT1 to GYF domain-containing proteins, such a connection might be achieved by a slightly different sequence because GYF domains may recognize PPG(F/I/L/M/V) motif as a general recognition signature (Kofler et al, 2007, Mol Cell Proteomics, DOI: <https://doi.org/10.1074/mcp.m500129-mcp200>). Indeed, certain URT1 homologs that do not contain a PPGF motif do have a PPG sequence followed by an hydrophobic

amino acid. We will address in a future study the possibility that URT1's PPGF motif is involved in the interaction with its GYF-domain partner proteins like EXA1.

3. Fig 2, with 8 replicates I would like to see error bars in a and b. Is Figure 2d an SDS gel? Please clarify.

The calculation of adjusted p-values takes into account the variance between the 8 replicates. The adjusted p-values are displayed on the Y-axis of the volcano plots shown in Fig. 2 and represent the reliability of this enrichment. Because we plot $-\log_{10}(\text{adjP})$, the higher a coordinate on the Y-axis, the more confident that this protein is enriched in IPs with the bait.

The legend for Figure 2d now states that "Pull-down and input fractions were analyzed by SDS-PAGE and SYPRO Ruby staining."

4. Figure 3, it is unclear why figures f and g are presented in 2 panels (one for rep 1-3, and one for 4-6, each). An average with error bars would be much more informative here. Same for Figure 4e, please change to mean with error bars, and figure 5. Or, for Figure 5, show one representative and move second to supplement. The figure is a bit confusing with that many bar graphs and having the rep2 moved to supplements should help simplify.

We agree with Reviewer 1 and revised the presentation of the 3'RACE-seq data. In all figures, the replicates are now plotted together in a single graph for each condition (e. g. one graph for polyadenylated tails in Ctrl, one graph for polyadenylated tails in URT1 samples, etc...). Instead of showing error bars, we show the average of all replicates (as a grey area) and the individual points for each replicate. This representation was chosen because we prefer to use a uniform presentation of 3'RACE-seq data throughout our study and Nature Communications guidelines require to plot individual data points rather than descriptive statistics for sample sizes $n < 10$.

5. Figure 4a, what is the half life of the RNA? While not necessary, it would be nice to have a number (error bars?) in addition to the gel.

The RNA substrate's half-lives were determined as described in Tang et al. 2019 Nat Struct Mol Biol DOI: 10.1038/s41594-019-0227-9 by quantifying the intact RNA substrate at each time point in three replicate experiments. Quantification and calculated half-lives are now shown as part of Fig. 5c. The corresponding quasi-Poisson regression curves are shown in Supplementary Fig. 5b and the raw data are available at <http://dx.doi.org/10.17632/ybcvmtcn9.2>.

Following a comment by Reviewer 3, we also included an additional experiment with an RNA substrate comprising a poly(U) tail.

Reviewer #2 (Remarks to the Author):

In this paper, the authors provide a thorough analysis of the factors that interact with the plant URT1 uridylyl transferase, and together with studies using mutant URT1 genes, describe a likely pathway by which Urt1 affects the half-life and pathway of degradation of the most polyadenylated mRNAs in plants. The authors have previously shown that URT1 uridylylates the oligoA tails (typically <25 nts), adding a small number of uridines, to the oligo(A) tails remaining after deadenylation. In many cases this is sufficient to restore a PABP binding site which could then stabilize those oligoadenylated mRNAs. Here they provide a much more detailed analysis of the overall role of URT1 in plant mRNA metabolism.

The most physiologically relevant data is the effect of URT1 expression on the polyA tails of 22 different Arabidopsis mRNAs which was done by comparing the null URT1 mutant with wild-type cells. These results confirm that a major role for uridylation by URT-1 is to allow accumulation of the "oligoadenylated" RNAs which are often uridylylated, and show that in the absence of uridylation there are many fewer of these "oligoadenylated" RNAs. Note that these data conflict with the idea that uridylation interacts with the decapping machinery to degrade RNA, although it is consistent with the idea that it acts to stabilize the oligoadenylated mRNAs, and possibly by recruiting these factors which inhibit translation, such as dhh1, prevent its translation. The data is consistent with URT-1 interacting with the deadenylation complex; however this interaction primarily may result in uridylyating and stabilizing oligoadenylated RNA, not promoting their degradation. The data also likely provide an explanation for why the GFP-reporter experiments resulted in constant mRNA but reduced protein.

We agree with many of these comments but we want to clarify one issue. Excessively deadenylated mRNAs (defined in the manuscript as having tails of less than 10 As) accumulate in absence of URT1 (Fig. 5). Therefore, URT1-mediated uridylation either prevents the formation of these excessively deadenylated mRNAs (by impeding deadenylation) or facilitates their turnover (by facilitating the recruitment of decapping activators). Indeed, we think that URT1 prevents the accumulation of excessively deadenylated mRNAs by both inhibiting excessive deadenylation and promoting 5'-3' degradation of deadenylated mRNAs.

We do not know yet whether the reconstitution of a binding site for PABP impacts mRNA stability (it could just favor the 5'-3' polarity of degradation) nor whether uridylation inhibits translation. The orthologs of several factors that are connected to URT1, such as Scd6 or GiGYF proteins indeed have a dual function in translational inhibition and activation of decapping. *Arabidopsis* DCP5 was also shown to be a translational inhibitor (at least *in vitro*). Therefore, we definitely do not exclude that the recruitment of DCP5 by URT1 could also impact translation. Yet,

our study does not demonstrate this aspect. In particular, we observe translation inhibition of *GFP* mRNAs upon overexpression of wild-type URT1, but also upon overexpression of URT1 with a mutated M1 motif which does not recruit DCP5 (see new Supplementary Fig. 3c and d). Therefore, further work is needed to elucidate whether and how URT1 impacts translation, but we think that this complex question should be addressed in a follow-up study.

The biological relevance of this pathway to preventing siRNA production which is detrimental to the organism is nicely shown and provides a rationale for the existence of this pathway in plants, and should also be indicated in their final figure. They might want to change the title to reflect these results.

We have modified our model to better integrate the role of URT1-mediated uridylation in preventing siRNA production. We also agree with the suggestion about the title, which was changed to “The TUTase URT1 connects decapping activators and prevents the accumulation of excessively deadenylated mRNAs to avoid siRNA biogenesis”.

Specific comments:

1. URT1 has a catalytic domain near the C-terminus and the N-terminal half of the protein is intrinsically disordered. They identify three small conserved motifs within this disordered region, that are conserved in a wide variety of plant species. It would be helpful to indicate the sequences of these motifs in Fig. 1 rather than in the supplemental figures, since it wasn't immediately clear that M1 and M2 are very different in sequence.

We fully agree that indicating M1 and M2 sequences in Figure 1 will be useful to readers. Figure 1 has been modified accordingly.

2. Using proteomics approaches they identify factors that interact with UTR1 (not necessarily direct binding partners, but potential complexes URT1 interacts with), using crosslinked samples. Here they find the CCR/NOT deadenylation complex as well as a number of known translational repressors and one decapping activator, DCP5 (Lsm14). Using the same approach without crosslinking provide a much simpler pattern, with DCP5, eIF4G and decapping activators including 3 ddx6 type RNA helicases.

3. They demonstrate that one of these proteins, DCP5, directly binds URT1 and that the M1 element in URT1 is required for binding, as is the LSm domain of DCP5. It would be helpful to comment on whether these two domains are sufficient for the interaction. Does mutating M2 have any effect?

Points 2 and 3 are both related to the direct URT1-DCP5 interaction and are addressed together. To test whether URT1's M1 motif and DCP5's LSm domain are sufficient for mediating the URT1-DCP5 interaction, we performed GST pull-down assays with the following recombinant proteins:

- 6His-MBP-DCP5^{LSm} corresponding to DCP5's LSm domain fused downstream of 6 histidines and MBP.
- 6His-GST-URT1¹⁻⁴⁰ corresponding to the 40 first aminoacids of URT1 (i.e. the N-terminal region of URT1 containing the M1 motif but not the M2 motif) fused downstream of 6 histidines and GST.
- 6His-GST-m1URT1¹⁻⁴⁰ corresponding to the 40 first aminoacids of URT1 in which the M1 motif is mutated (L21N, L25N) fused downstream of 6 histidines and GST.

These GST pull-down assays detected an interaction between the 40 N-terminal amino-acids of URT1 and the LSm domain of DCP5 that was lost upon mutating URT1 M1 motif. Because only a small amount of 6His-MBP-DCP5^{LSm} was pulled down by 6His-GST-URT1¹⁻⁴⁰, it is possible that additional regions of URT1 and/or DCP5 contribute to their interaction. An alternative explanation could be that the interaction is hampered by poor folding of the short protein fragments fused to tags. Nevertheless, these data provided as Supplementary Figure 2b indicate that the URT1 M1 motif and DCP5 LSm motif are sufficient for the URT1-DCP5 interaction.

Additional GST pull-down assays showed that mutating URT1 M2 motif has no effect on the URT1-DCP5 interaction *in vitro*. These data are presented in the new Supplementary Figure 2c.

4. The rest of the paper is focused on changes that occur on the polyA tail of a GFP reporter mRNAs that result from expression of URT1 and various URT1 mutants in the presence of the endogenous tobacco uridyl transferases. Expression of active UTR1 reduces GFP protein expression, while catalytically inactive URT1 had no effect. However expression of active URT1 had no effect on GFP mRNA levels. Active URT1 expression resulted in increased uridylation, with uridylation of long polyA tails as well as “oligoadenylated” tails. Overall there was an increased amount of long polyA tails. A similar effect was seen on an endogenous mRNA, PR2 mRNA. This non-physiological result (presumably requires overexpression of uridyltransferases), uncovered an effect of uridylation on long polyA tails which resulted in the decreased deadenylation, likely explained by the biochemical properties of the CCR4 enzyme.

In the revised version of this manuscript we determined global mRNA poly(A) tail profiles in wild-type and *urt1 Arabidopsis* plants by nanopore direct RNA sequencing. These data, presented in the new Fig. 5a, confirm at a transcriptome-wide level that loss of URT1 has a deep impact on poly(A) tail profiles. In particular, we observe a decrease in poly(A) tail size for the majority of the *Arabidopsis* mRNAs. These data provide strong additional support for our conclusion that URT1 has a key role in controlling poly(A) sizes.

5. The role of the M1 region, which interacts with the dcp5 factor, could be to recruit the factors, that prevent translation of the mRNA (ddx6 [dhh1], pat1) and actually help stabilize the oligoadenylated mRNA. This would be similar to the accumulation of deadenylated mRNA after stress in yeast, shown by Collier and Parker, providing a pool of mRNA which could then be readenylated for future translation.

We fully agree that this hypothesis is plausible but it is not experimentally proven yet. As mentioned above, the translational inhibition of *GFP* mRNAs induced by URT1 overexpression is not abrogated by mutating the M1 motif. At present, our data do not demonstrate that the M1 motif mediates translational inhibition by URT1. Yet it could definitely participate to such a process. We are currently addressing the potential involvement of the additional SLiMs in translational inhibition, to test their possible redundancy in recruiting translational repressors.

6. I am unclear about the origin of the “A-rich tails”. It seems unlikely that they come from inaccurate polyadenylation. More likely they may come from incomplete deadenylation, followed by incorporation of a non-templated nucleotide and then cytoplasmic adenylation.

We also think that this scenario should be tested, but we have no experimental evidence yet to support this working hypothesis. We think that the experimental conditions of the *in planta* assay described in our study create an opportunity to experimentally address both the biogenesis and functions of these tails.

7. The final model figure needs to show more clearly that a major role of uridylation is to prevent the unregulated degradation of oligoadenylated RNA, which can result in siRNA production. The tight coupling of uridylation to deadenylation by interacting with the NOT1/CCR4 complex may primarily serve to do this, and not to direct complete degradation of the mRNA.

We fully agree with the comment on the primary biological function of URT1-mediated uridylation. We have added a panel to further highlight this important function of URT1. As explained above, we also agreed with Reviewer 2's suggestion to modify the title to reflect this point.

Reviewer #3 (Remarks to the Author):

In this manuscript, Scheer et al, report the connection between the TUTase URT1 and the decapping activators. The authors found that URT1 contain a conserved N-terminal intrinsically disordered region (IDR) and identified two short linear motifs (SLiMs) . Then the authors identified potential interaction proteins of URT1 through mass-spectrometry analyses of purified URT1 complex. The authors showed that SLiM1 mediates the interaction between DCP5 and URT1. The authors further show transient expression of URT1 can increase the ratio long poly-A containing GFP mRNAs gene in *Nicotiana benthamiana* leaves. The authors showed 3' terminal 1 or 2 Us can prevent the degradation of poly-A tails by CAF1b in vitro. The authors propose that ectopic expression of URT1 may trigger the degradation of oligoadenylated GFP mRNAs or prevent deadenylation of mRNAs, and thereby change the poly A profiles. The authors show that transient Expression an inactive UTR1D491/3A increased the accumulation oligoadenylated (16-25 nt A) GFP mRNAs with U-tails, which depends on the SLiM1 domain. The authors suggest that UTR1D491/3A may have a dominant negative impact on endogenous URT1 of *Nicotiana benthamiana* by depleting factors degrading uridylated oligoadenylated mRNA through its SLiM1 domain. The authors further show that in *Arabidopsis* *urt1* mutant, a slight shift toward to short poly A tails occurs. The authors further show that absence of URT1 and XRN4 can trigger the production of siRNAs.

Concerns are:

1- The manuscript describes several different aspects of URT1. Each of them seems to be incomplete story and need additional evidences to support the conclusion.

We report three important novel findings and we think that all three conclusions are demonstrated in the revised version: (i) a TUTase interacts with decapping activators, (ii) uridylation shapes poly(A) tails by preventing the accumulation of excessively deadenylated mRNAs and (iii) URT1-mediated uridylation prevents the biogenesis of spurious siRNAs targeting mRNAs. In the revised version, our conclusions gain further support from the analysis of mRNA poly(A) tails by nanopore direct RNA sequencing. This new unbiased experiment demonstrates a global reduction of poly(A) tail length upon loss of URT1 and confirms the overaccumulation of excessively deadenylated mRNAs (new Fig. 5a). Taken together, these data unequivocally demonstrate the key role of URT1 in shaping poly(A) tails. We detailed all evidence supporting our conclusions in our point-by-point answers below.

2- The claim that URT1 may trigger the degradation of oligoadenylated GFP mRNAs or prevent deadenylation of mRNAs need support from experimental evidences besides speculations from sequencing data.

To address this comment, we have added new data and re-organized the manuscript to put the emphasis on the key finding that URT1-mediated uridylation shapes poly(A) tails by preventing the accumulation of excessively deadenylated mRNAs.

Our strategy to investigate URT1's functions is to first use the *N. benthamiana* leaf patch assay to exacerbate and identify key roles of URT1-mediated uridylation, and then to validate these observations in *Arabidopsis*. The main information gained from the reporter mRNA analysis in *N. benthamiana* is that overexpression of URT1 leads to a decrease of oligoadenylated mRNAs, and to an increase of mRNA tail size. If those effects reflect *bona fide* functions of URT1, we should observe the exact opposite phenotype in *urt1 Arabidopsis* mutants. And this is what we observed for the 22 mRNAs analyzed in the first version of the manuscript. In the revised version, we measured mRNA poly(A) tails by nanopore direct RNA sequencing of both wild-type and mutants *Arabidopsis* plants. These new data confirm the global reduction of poly(A) tail length and the accumulation of excessively deadenylated mRNAs upon loss of URT1 (new Fig. 5a). Altogether our results leave no doubts that mRNA uridylation shapes poly(A) tails in *Arabidopsis*. We think that the impact of URT1 in shaping poly(A) tails is a key information, and that it is now demonstrated unequivocally.

We have reorganized the presentation of the results to make it clear that the role of URT1 in shaping poly(A) tails in *Arabidopsis* represents the key finding of this part of the manuscript. Nevertheless, we propose two non-mutually exclusive mechanisms by which URT1 impacts poly(A) tail profiles, and both are supported by our data.

1: URT1 preferentially targets oligoadenylated mRNAs and accelerates their degradation.

In Sement et al. Nucleic Acids Res 2013, Zuber et al. Cell Reports 2016 and the present manuscript, we consistently observe that URT1 preferentially uridylates deadenylated mRNAs in *Arabidopsis*. We consider this preference of URT1 for deadenylated mRNAs as a demonstrated fact. We have also previously shown that decapped *LOM1* mRNAs are uridylated up to 90 % and have a median oligo(A) size of 11, whereas capped *LOM1* mRNAs are uridylated to 1% and have a median poly(A) tail size of 50 nucleotides. Decapped and uridylated mRNAs were detected only in *xrn4* mutants, in which cytosolic 5'-3' degradation is compromised. These data already supported a link between uridylation and decay of deadenylated mRNAs, similarly to what the Norbury lab originally reported in *S. pombe*. This conclusion is in agreement with the primordial role of mRNA uridylation, conserved across eukaryotes. However, no decisive experiment proving that uridylation accelerates the degradation of oligoadenylated mRNAs in plants has been reported yet. There are many different possibilities to explain this deficiency. One is likely due to the redundancy of decay pathways in plants. For instance, current data point to a multiplicity of molecular interactions inducing decapping. Also, a global relationship between uridylation and decay remains to be reported in *Arabidopsis*. In the revised version, we now reveal a global relationship between uridylation and mRNA decay. We took advantage of the depth of our TAIL-seq data to calculate mRNA uridylation rates. We then compared these uridylation rates with three independent datasets on mRNA half-lives in *Arabidopsis* (Narsai et al. Plant Cell 2007; Sorenson et al. PNAS 2018; Szabo et al. Plant Cell 2020). We find that the higher the propension of an mRNA to get uridylated, the shorter its half-life. These new data, presented in Fig. 5d, robustly link uridylation to the general process of mRNA decay on a transcriptome-wide level.

2: Uridylation intrinsically impedes deadenylation.

Despite the preference of URT1 for uridylating deadenylated mRNAs, a basal level of uridylation of long tails is detected both in control *N. benthamiana* and wild-type *Arabidopsis* plants as shown in Fig. 3f and S5a, respectively. Therefore, the uridylation of long poly(A) tails is naturally happening but at a rather low occurrence. This low frequency could be explained by the competition of diverse factors including deadenylases, PABP and TUTases. to access the 3' extremity of poly(A) tails. Overexpressing URT1 likely favor its access to poly(A) tails, resulting in increased uridylation levels (Fig. 3f). Yet, the addition and removal of uridines must be a dynamic process (e.g. antagonistic action between URT1 and 3'-5' exoribonucleases, including deadenylases). Therefore, a mixture of long poly(A) tails with and without terminal uridines accumulate upon overexpressing URT1. This interpretation is fully supported by our biochemical assays which demonstrate that a single uridine slows down a deadenylase of the CAF1 family, CAF1b (now in Fig. 5c). Two uridines further slow down deadenylation, though not fully abrogate degradation, demonstrating that deadenylation can overcome the terminal uridines, provided they are only few. Yet, as we show now in the revised version, longer homopolymeric U-tails strongly inhibit CAF1 *in vitro* (see also answer to point 9). Our observations are in agreement with recent data obtained with the reconstituted CCR4-NOT complex from *S. pombe* (Tang et al. NSMB 2019) and support the conclusion that uridylation could intrinsically impede deadenylation.

Although we certainly agree that future work is needed to fully elucidate all molecular mechanisms by which URT1 influences poly(A) tail length and mRNA stability, we would like to point out that both *in planta* and *in vitro* data support the two mechanisms that we propose for URT1 action. More importantly, the key result that URT1 shapes poly(A) tail size, is now demonstrated in *Arabidopsis* at a global scale by nanopore DRS. To state more clearly that this is this main conclusion of our study, we have reorganized the presentation of the data and the text.

3- The majority of experiments were transient overexpression experiments, which may not reflect the normal physiological condition.

We agree with Reviewer 3 that transient overexpression experiments do not reflect normal physiological condition. However, most information in biology is deduced from experiments using single or multiple mutants, which also do not represent "normal physiological condition". Countless information has also been obtained from *in vitro* assays

or using cell lines grown *in vitro*. Finally, much of our mechanistic insight in conserved RNA decay processes originates from reporter studies in yeast. The transient expression of transgenes in *N. benthamiana* leaves is a comparable classical assay, and was successfully employed to elucidate key aspects of e.g. RNA silencing, nonsense-mediated decay and many other basic mechanisms.

Here we use this assay to determine basic consequences of mRNA uridylation on poly(A) tail length. The insight gained through this technique was then validated in wild-type *Arabidopsis* plants. Importantly, we now provide new nanopore direct RNA sequencing data that confirm the impact of URT1-mediated uridylation in shaping poly(A) tails in *Arabidopsis*. Indeed, the nanopore data demonstrate at a transcriptome-wide level that the lack of URT1 results in the accumulation of mRNAs with short oligoadenylated tails in *Arabidopsis*, whereas long poly(A) tails are decreased (new Fig. 5a). Therefore, we are convinced that URT1-mediated uridylation shapes poly(A) tail profiles, which is a novel and important information on the roles of RNA uridylation.

4- Is IDR also conserved in other organisms or specific to plants

The presence of an IDR in TUTases is not specific to plants. However, position, composition and length are not conserved. For instance, a potential IDR of the TUTase Cid1 from *S. pombe* is an extremely short C-terminal sequence, whereas the IDRs of plant URT1 proteins are ca 45 kDa and at their N-termini. The human TUTases TUT4 and TUT7 have 3 IDRs in N-terminal, middle and C-terminal positions. We have reviewed this aspect in a dedicated part of Scheer et al. Trends Genet 2016. We would prefer not to discuss this aspect in the first paragraph of the results as it would disrupt the flow.

5- Page 3, “Poales, which include key cereal crops, are among the plants whose genome encodes two URT1 paralogs, URT1A and URT1B” Paralog means homologs with different functions. Are the authors sure that URT1A and URT1b have different functions with URT1?

Paralogs are homologous genes that evolved by gene duplication in a species. Paralogy does not necessarily imply different functions, although this is frequently observed. To avoid any misunderstanding, the sentence was changed to: “Poales, which include key cereal crops, are among the plants whose genome encodes two URT1 homologous sequences, URT1A and URT1B”.

The use of “paralog” was also changed to “homologs” when we mention DCP5 and DCP5L, and VCS and VCR, respectively.

6- All the MASS experiments were performed using overexpression lines. The authors should confirm the interaction under native conditions using native promoters; at least show URT1 indeed interacts with DCP5 under native condition (native promoters).

Indeed, overexpression of a bait may eventually lead to artefactual interactions. Yet, overexpression can also enhance the formation of specific but low-affinity complexes between partners. The important point when using overexpressing lines is to validate the detected molecular interaction. In that respect, we are confident that we identified a direct interaction between URT1 and DCP5 because of all the following results that must be considered altogether:

- DCP5 is the most significantly and the most enriched protein co-purifying with URT1 using IPs with no cross-link (Fig. 2b)
- DCP5 and URT1 directly interact *in vitro* (Fig. 2d)
- a SLiM (Motif 1) conserved across land plant evolution (Fig. 1d and 1e) is necessary for the URT1-DCP5 interaction *in vitro* (Fig. 2d). Similarly, the LSm domain of DCP5 is required for the URT1-DCP5 interaction *in vitro* (Fig. 2d).
- URT1 M1 motif and DCP5 LSm domain are sufficient for the interaction (new Supplementary Fig. 2b).
- An interaction between a Leucine-rich motif resembling M1 and a LSm domain is conserved in yeasts and humans to mediate the interaction between the DCP5 ortholog and decapping factors.
- lastly and most importantly, the most significantly depleted protein when comparing IPs with wild-type URT1 and URT1 mutated in the M1 motif is DCP5 (and DCP5-like) (Fig 2f). This result demonstrates that even if URT1 is overexpressed in planta, the mutations of two aminoacids disrupt the URT1-DCP5 interaction.

To further support our conclusion and to specifically address this comment by Reviewer 3, we performed DCP5 IPs using lines expressing DCP5-GFP at endogenous level. The eluates were probed with anti-URT1 antibodies and this western blot analysis also confirmed the co-purification of DCP5 and URT1 when both proteins are expressed at endogenous levels. Those new data are now presented in the new Supplementary Fig. 2a. We think that, taken altogether, our data prove beyond doubts the direct interaction between URT1 and DCP5.

7- Page 5, Please provide explanation of the shift towards longer poly(A) tails without U-tail induced by URT1 expression. Two possibilities provided here cannot explain this observation.

As stated above, we propose two mechanisms by which URT1-mediated uridylation could shape poly(A) tails: by favoring the degradation of oligoadenylated mRNAs and by impeding deadenylation. These two roles are not mutually exclusive and can both contribute to explain why URT1 overexpression induces a shift towards longer poly(A) tails that are not uridylated. Both roles are supported by experimental data as detailed in our answer to comments 2 and 12.

Role 1: URT1 targets preferentially oligoadenylated mRNAs and accelerates their degradation, thereby reducing the pool of mRNAs with oligo(A) tails (see detailed response to comment 2). Because poly(A) profiles are presented as a distribution of poly(A) tail sizes (i.e. the added proportions of all tail sizes will equal 1), a reduction in the proportion of mRNAs with oligo(A) tails will increase the proportion of mRNAs with long tails (i.e. without 3' terminal uridines).

Role 2: Uridylation intrinsically impedes deadenylation. Overexpressing URT1 in *N. benthamiana* dramatically increases the uridylation of longer tails as compared to the wild-type situation. Because the addition of 3' terminal uridines delays but not completely abrogates deadenylation, the addition and removal of terminal uridines is a dynamic process that results from the competing actions of various factors (see our answer to comment 2). Therefore, both long poly(A) tails with and without uridines accumulate.

To take comment 7 into consideration, we have extensively modified the text to better explain how URT1 can influence poly(A) tail distribution.

8- Figure 3, what happens to the GFP uridylation when URT1P618L is expressed. This one should give additional confirmation for the result since it has a weak activity and may have intermediate impact on GFP uridylation.

Reviewer 3 is right. In fact, URT1P618L samples for replicates 1 and 2 were analyzed by 3'RACE-seq alongside the other samples. As compared with the patterns observed in URT1 and URT1^{D491/3A}, URT1P618L samples indeed show an intermediate pattern for uridylated mRNAs (See Fig. 1 for Reviewers). This observation is perfectly in line with the weak activity of URT1P618L. Because URT1P618L has this residual intermediate activity between wild-type URT1 and the catalytic inactive mutant, the URT1P618L protein also accumulates to intermediate levels between URT1 and URT1^{D491/3A} (Fig. 3b). Both the residual activity and the intermediate level of accumulation complexify the presentation of the 3'RACE-seq data without revealing any fundamental novel information. Therefore, we prefer not to include these data in the manuscript.

9- The authors examined the impact of 1 or 2 U on CAF1. What is impact of long u-tail on CAF1 activity? Can URT1 add long U tails?

To answer this comment, we tested the impact of a U14 tail on the activity of CAF1b. As expected, this tail strongly inhibits CAF1b (see new Fig. 5c). Of note, most mRNAs (~98%) are tailed with 1 to 3 Us as determined by TAIL-seq or 3'RACE-seq (e. g. see Supplementary Fig. 3f). Of course, we cannot exclude that longer U-tails exist *in vivo* and that they escape detection because of extremely low abundance or technical issues. Yet, there is no evidence to date that tailing mRNAs with such long poly(U) tails has a biological relevance.

10- Figure 3 and 4, How was the size distribution of u-tailed or non-u-tailed RNAs calculated? Using total readings or using total U-tailed RNA or non- U-tailed RNAs readings to calculate the ratios of various sizes? It seems the total readings were used. If so, the observations may be due to the ratio changes of uridylated populations. It will nice to have a size distribution in u-tailed or non-u tail RNA populations

Indeed the size distribution of U-tailed or non-U-tailed mRNAs was normalized against the total number of sequences. We recalculated the size distributions of U tailed or non-U-tailed RNAs as percentages of the total number of U tailed or non-U-tailed RNAs, respectively (see Fig. 2 for Reviewers). We still observe the increased accumulation of long poly(A) tails for both U tailed or non-U-tailed mRNAs with this alternative normalization. Therefore, this main observation is not due to the ratio changes of uridylated populations. Yet, by using the total number of U-tailed mRNAs for normalization, we lose the information regarding the accumulation of U-tailed RNA population in URT1^{D491/3A}. Therefore, we decided to maintain the normalization against the total number of sequences in Fig. 3, Fig. 4 and Supplementary Fig 4. The information "The percentages were calculated using the total number of sequences with tails from 1 to 90 nucleotides." was added to the Methods section and the figure legends. By contrast, Supplementary Fig. 5 shows the size distribution of uridylated reads normalized against all uridylated sequences to facilitate the comparison between the 22 *Arabidopsis* mRNAs. This is now clearly stated in the legend.

11- Figure 3e, the ratios of U-tailed GFP mRNAs in different UTR1D491/3A replicate (1, 2,3; vs 4, 5, 6) varies. Any explanation on this?

We do not have a specific explanation for the slight increased uridylation levels observed in rep 4,5,6 vs 1,2,3. Because we detect this slight increase in 3 biological replicates analyzed by a single MiSeq run, we cannot exclude a technical issue. For example, we observed that a clustering issue on the MiSeq flow-cell may have favored the detection of shorter tails in some runs. Importantly, this increase is marginal and we do not draw any conclusion from it. The important conclusions supported by statistical analyses shown in Figure 3e are that (i) uridylation of *GFP* mRNAs is significantly increased upon expression of URT1 and (ii) that uridylation upon expression of URT1^{D491/3A} is intermediate between control and URT1 samples. This intermediate increase depends on the M1 motif as shown in Fig. 4c.

12- The claim UTR1D491/3A triggers mRNA degradation is weak, as no evidence to show u-tailed GFP and PR2 mRNAs decay faster than controls. Similar amount of GFP mRNAs were observed (Figure 3). Moreover, there is no evidence to show that URT1 depletes degradation components. The increased PR2 levels could

be caused by different inducing levels. mRNA decay experiment is needed to examine the effect of URT1 on mRNA stability.

To simplify our answers, we have split this comment in three points that we address sequentially.

- 12a The claim UTR1D491/3A triggers mRNA degradation is weak, as no evidence to show u-tailed GFP and PR2 mRNAs decay faster than controls. mRNA decay experiment is needed to examine the effect of URT1 on mRNA stability.

We read the comment as we should demonstrate that URT1 (not URT1^{D491/3A}) triggers mRNA degradation. We attempted to measure mRNA decay rate in infiltrated *N. benthamiana* leaves by trying to block transcription with actinomycin D or cordycepin. Yet, we still observed some erratic transcriptional activity in some treated samples, notably at the beginning of the time-course. The data that we obtained with this approach are highly variable and are therefore not translatable in meaningful results. We decided to drop this approach, also because the *N. benthamiana* transient expression assay is mainly used to analyze GFP mRNA poly(A) tail profiles and draw working hypotheses. Yet, the goal is to understand the impact of uridylation on mRNA metabolism in wild-type *Arabidopsis*. We therefore chose to address this point using global datasets generated in *Arabidopsis*. We used our TAIL-seq data to classify mRNAs according to their uridylation rate. We then compared the uridylation status of these mRNAs with each of the three available datasets of *Arabidopsis* mRNA half-lives. Please note that each of these studies used a different tissue and a distinct protocol (transcription inhibition by actinomycin D or by cordycepin, and RNA metabolic labeling). These differences may partially explain why half-lives that were calculated for individual mRNAs differ remarkably among the three datasets (Szabo et al. Plant Cell 2020). Other explanations may include indirect effects caused by transcription inhibition or the uptake of 5-ethynyl uridine, the induction of deadenylases by the vacuum infiltration necessary for drug penetration or the likely impact of the ATP analog cordycepin on poly(A) tail synthesis and stability. Yet, despite the lack of correlation between the three datasets, we find a robust relationship between uridylation levels and mRNA half-lives. This analysis is shown in Fig. 5d and reveals that *Arabidopsis* mRNAs that are preferentially uridylated have significantly lower half-lives. We are aware that this approach does not solve mechanistic details, that will be addressed in future work. Also, this observation was expectable given the conservation of uridylation main function across eukaryotes. Yet, it represents the first report of a correlation between mRNA uridylation and decay at transcriptome-wide level in *Arabidopsis*.

- 12b there is no evidence to show that URT1 depletes degradation components.

First of all, we want to clarify our hypothesis. By “deplete at least one factor involved in the turnover of uridylated oligoadenylated mRNAs through an interaction involving the M1 motif”, we meant that such factors are depleted from the available pool. We now use the word “sequester” rather than “deplete” in the text to avoid misunderstanding. Secondly, we think that URT1^{D491/3A} (not URT1) sequesters, due to its high overexpression, at least one factor involved in RNA decay. This is not the case for active URT1 because it does not accumulate to such high levels. Because of the conservation of the M1 motif across plant URT1 orthologs, we hypothesized that this decay factor could be NbDCP5. To test this idea, we have now investigated the *N. benthamiana* interactome of AtURT1^{D491/3A} and At-m1URT1^{D491/3A} by mass spectrometry. Those results confirmed our hypothesis by showing that AtURT1^{D491/3A} can interact with *N. benthamiana* DCP5 isoforms (and additional decay factors like orthologs of RH6-8-10 RNA helicases), and that this interaction is lost by mutating the M1 motif (even though At-m1URT1^{D491/3A} is overexpressed at high levels). The interaction of AtURT1^{D491/3A} with *N. benthamiana* decay factors strongly supports our hypothesis that the large overexpression of AtURT1^{D491/3A} sequesters decay factors including NbDCP5, thereby inhibiting RNA degradation. The new MS data are presented in Supplementary Fig. 3c-d.

- 12c The increased PR2 levels could be caused by different inducing levels

Our new mass spectrometry data show that AtURT1^{D491/3A} (but not At-m1URT1^{D491/3A}) interacts with NbDCP5 and other decay factors. We therefore favor the idea that the large overexpression of AtURT1^{D491/3A} sequesters decay factors, thereby impairing RNA degradation. However, we definitely agree that we cannot exclude that overexpression of At-URT1^{D491/3A} systematically induces PR2 transcription. Therefore, the text was changed to: “overexpression of URT1^{D491/3A} led to increased levels of PR2 mRNAs (Supplementary Fig. 4a). Either overexpression of URT1^{D491/3A} induces PR2 transcription at a higher level, or URT1^{D491/3A} overexpression impairs PR2 mRNA turnover”. These hypotheses are now proposed before presenting the 3'RACE-seq data on PR2 mRNAs because the main result of this paragraph is about the similarities in poly(A) profiles observed for GFP and PR2 mRNAs.

REVIEWERS' COMMENTS

Reviewer #1 (Remarks to the Author):

The authors have sufficiently addressed my comments in the revision.

Reviewer #2 (Remarks to the Author):

The authors have responded thoroughly to the initial review, both making the message of the paper clearer as well as adding substantial additional data to support their model. A major strength of the paper is the combination of biochemistry with genetics, and novel sequencing approaches. In particular, they now show clearly that one role of the TUTase is to accelerate removal of the short oligo(A) tails, which prevents unwanted siRNA production. They clearly distinguish between the effects on the reporter gene, and on the endogenous mRNAs in Arabidopsis.

Specific comments:

1. The authors make the point that the mutant changes the distribution of polyA tail length, which is correct. However, this effect is a result of the accumulation of many shorter A tails (whose mRNAs are normally degraded). Thus the effect is due to the failure of this degradation step rather than an effect on distribution of the length of "functional" polyA tails, since the mRNAs with short tails are not translated. They might make the effect clearer, and point out that there is no effect on the distribution of the bulk of the polyA tails. In particular on the bottom of page 6, lines 305-306 that there is a "robust effect" on polyA tail sizes" is somewhat misleading.
2. In the discussion of the role of uridylation on the short "deadenylated" mRNAs, [page 9, l. 440] they might mention the recent paper (Montemayor et al., 2020) which shows that unlike *S. cerevisiae*, the Lsm 1-7 of *S. pombe* (and suggests that the same is true in many other eucaryotes), that Lsm 1-7 does not bind short A tails directly but greatly prefers tails with U in them. This provides a potential mechanism for uridylation promoting recruitment of Lsm1-7.
3. The finding that there are mixed tails after expression of the mutant URT1 is surprising; one way these tails could arise, is by deadenylation followed by uridylation and then cytoplasmic polyadenylation; they don't have to arise during the initial A tail formation (Ochi and Chiba, 2016).

Minor:

1. Some minor editing changes: l. 135, page 3 "coherent" should be "consistent"; l. 253 pg 5 "more expressed" should be "expressed more highly".

Montemayor, E. J., Virta, J. M., Hayes, S. M., Nomura, Y., Brow, D. A. & Butcher, S. E. (2020). Molecular basis for the distinct cellular functions of the Lsm1-7 and Lsm2-8 complexes. *RNA*, 26, 1400-1413.

Ochi, H. & Chiba, K. (2016). Hormonal stimulation of starfish oocytes induces partial degradation of the 3' termini of cyclin B mRNAs with oligo(U) tails, followed by poly(A) elongation. *RNA*, 22, 822-829.

Reviewer #3 (Remarks to the Author):

My concerns have been addressed.

ANSWERS TO REVIEWERS' COMMENTS

Reviewer #1 (Remarks to the Author):

The authors have sufficiently addressed my comments in the revision.

Reviewer #2 (Remarks to the Author):

The authors have responded thoroughly to the initial review, both making the message of the paper clearer as well as adding substantial additional data to support their model. A major strength of the paper is the combination of biochemistry with genetics, and novel sequencing approaches. In particular, they now show clearly that one role of the TUTase is to accelerate removal of the short oligo(A) tails, which prevents unwanted siRNA production. They clearly distinguish between the effects on the reporter gene, and on the endogenous mRNAs in Arabidopsis.

Specific comments:

1. The authors make the point that the mutant changes the distribution of polyA tail length, which is correct. However, this effect is a result of the accumulation of many shorter A tails (whose mRNAs are normally degraded). Thus the effect is due to the failure of this degradation step rather than an effect on distribution of the length of “functional” polyA tails, since the mRNAs with short tails are not translated. They might make the effect clearer, and point out that there is no effect on the distribution of the bulk of the polyA tails. In particular on the bottom of page 6, lines 305-306 that there is a “robust effect” on polyA tail sizes” is somewhat misleading.

We definitely agree that the distribution of polyA tail length is influenced by the accumulation of mRNA with shorter A tails in the *urt1* mutant. However, we cannot exclude at this stage that URT1 has also an effect on the length of longer polyA tails (e.g. protection against deadenylation). To address this comment of Reviewer 2 and to avoid a potential misunderstanding by readers, we rephrased the description of the Nanopore results. Yet, we would prefer to keep the word “robust” because we think that it is a factual description of URT1’s impact on mRNA poly(A) tail distribution. The new paragraph is now: “The nanopore DRS analysis revealed the impact of URT1 on poly(A) tail size distributions with a clear shift of the distribution towards short oligo(A) tails in *urt1* samples (Fig. 5a). This accumulation of deadenylated mRNAs is observed for both bulk and intergenic poly(A) tail size distributions, thereby reflecting a robust effect.”

2. In the discussion of the role of uridylation on the short “deadenylated” mRNAs, [page 9, l. 440] they might mention the recent paper (Montemayor et al., 2020) which shows that unlike *S.cerevisiae*, the Lsm 1-7 of *S. pombe* (and suggests that the same is true in many other eucaryotes), that Lsm 1-7 does not bind short A tails directly but greatly prefers tails with U in them. This provides a potential mechanism for uridylation promoting recruitment of Lsm1-7.

We thank Reviewer 2 for the suggestion. The Montemayor et al. paper is now cited. We also cite Lobel *et al.* that reports that Pat1 activates binding of LSm1-7 to oligoA tails.

Lobel, J. H., Tibble, R. W. & Gross, J. D. Pat1 activates late steps in mRNA decay by multiple mechanisms. *Proc. Natl. Acad. Sci.* **116**, 23512–23517 (2019).

3. The finding that there are mixed tails after expression of the mutant URT1 is surprising; one way these tails could arise, is by deadenylation followed by uridylation and then cytoplasmic polyadenylation; they don’t have to arise during the initial A tail formation (Ochi and Chiba, 2016).

We definitely agree that the A-rich tails are likely added in the cytosol after the initial poly(A) tail has been shortened. However, we do not investigate their synthesis in the present manuscript and therefore we prefer not to speculate on this point.

Minor:

1. Some minor editing changes: l. 135, page 3 “coherent” should be “consistent”; l. 253 pg 5 “more expressed” should be “expressed more highly”.

Corrected.

Reviewer #3 (Remarks to the Author):

My concerns have been addressed.